# Lem2 and Lnp1 maintain the membrane boundary between the nuclear envelope and endoplasmic reticulum

Yasuhiro Hirano [1✉], Yasuha Kinugasa [1,4], Hiroko Osakada[2], Tomoko Shindo[3], Yoshino Kubota[1], Shinsuke Shibata[3], Tokuko Haraguchi [1,2] & Yasushi Hiraoka [1,2✉]

The nuclear envelope (NE) continues to the endoplasmic reticulum (ER). Proper partitioning of NE and ER is crucial for cellular activity, but the key factors maintaining the boundary between NE and ER remain to be elucidated. Here we show that the conserved membrane proteins Lem2 and Lnp1 cooperatively play a crucial role in maintaining the NE-ER membrane boundary in fission yeast *Schizosaccharomyces pombe*. Cells lacking both Lem2 and Lnp1 caused severe growth defects associated with aberrant expansion of the NE/ER membranes, abnormal leakage of nuclear proteins, and abnormal formation of vacuolar-like structures in the nucleus. Overexpression of the ER membrane protein Apq12 rescued the growth defect associated with membrane disorder caused by the loss of Lem2 and Lnp1. Genetic analysis showed that Apq12 had overlapping functions with Lnp1. We propose that a membrane protein network with Lem2 and Lnp1 acts as a critical factor to maintain the NE-ER boundary.

[1] Graduate School of Frontier Biosciences, Osaka University, Suita 565-0871, Japan. [2] Advanced ICT Research Institute Kobe, National Institute of Information and Communications Technology, Kobe 651-2492, Japan. [3] Electron Microscope Laboratory, Keio University School of Medicine, Shinjuku-ku, Tokyo 160-8582, Japan. [4]Present address: Research Institute for Radiation Biology and Medicine, Hiroshima University, Hiroshima 734-8553, Japan. ✉email: yhira@fbs.osaka-u.ac.jp; hiraoka@fbs.osaka-u.ac.jp

The eukaryotic genome is organized within the nucleus surrounded by the nuclear envelope (NE). The NE comprises of inner and outer nuclear membranes, which are continuous with the endoplasmic reticulum (ER) membranes. Integrity of the NE is crucial for retaining the nuclear barrier and thereby maintaining cell viability[1]; yet, the NE is dynamic, repeatedly breaking down and reforming during mitosis (open mitosis) in higher eukaryotic cells[2,3].

To seal the membrane holes during NE reformation at the end of mitosis, the endosomal sorting complex required for transport-III (ESCRT-III) is transiently recruited to the reforming NE by charged multivesicular body protein 7 (CHMP7 in metazoa; Chm7 in the budding yeast *Saccharomyces cerevisiae*, and Cmp7 in the fission yeast *Schizosaccharomyces pombe*)[4–8]. ATPase associated with diverse cellular activities (AAA)-ATPase Vps4 is loaded on the NE reforming sites and induces disassembly of ESCRT-III to complete the NE sealing[5,8,9]. Recent studies have demonstrated that Lem2, a member of the conserved LEM-domain NE protein family, is also involved in the nuclear membrane sealing at the end of mitosis[5–8,10,11] by acting as a nuclear-targeting adaptor for CHMP7[5].

Unlike "open mitosis" in metazoan cells, fungal cells undergo "closed mitosis," in which the NE remains intact throughout the process of mitosis[12–14]. In *S. pombe*, however, the spindle pole body (SPB) is inserted into the NE prior to mitosis and excluded from the NE at the end of mitosis and thus requires sealing of the hole of the NE[15,16]. All ESCRT-III components and related proteins are conserved in *S. pombe*[5], suggesting that *S. pombe* may also be using similar mechanisms to seal the membrane hole. ESCRT-III is involved in sealing a membrane rupture not only during mitosis but also during interphase in mammalian cells, illustrating its global role in maintaining NE integrity[17,18]. *S. pombe* Lem2, which shares LEM-like domain (HEH/LEM domain) with metazoan Lem2, plays a role in the integrity of the NE and in stability of the chromosomes[5,19–25]. Defects in the cells lacking Lem2 are suppressed by an increased amount of Lnp1[24]. Lnp1 (homolog of human Lunapark) is an ER membrane protein conserved from yeasts to humans. Lnp1 localizes at the three-way junction of the tubular ER network in humans and *S. cerevisiae* and stabilizes the junction in cooperation with GTPase Sey1 (a yeast homolog of human atlastin that mediates ER membrane fusion) and reticulons and DP1/Yop1p (curvature-stabilizing proteins)[26–31]. Accordingly, loss of Lem2 and Lnp1 induces severe growth defects[24], suggesting that these proteins have a cooperative role in crucial cellular processes. However, the roles of these NE/ER membrane proteins in cellular processes and the functional relationship between these membrane proteins remain unknown.

To understand the roles of the NE/ER membrane proteins in cellular processes, we observed the morphology of the intracellular membranes in *S. pombe* cells lacking these proteins using fluorescence microscopy and electron microscopy. To further understand the functional relationship between several membrane proteins, genetic interaction analyses were performed in combination with cytological analyses. In this report, we describe the role of Lem2 and Lnp1 in maintaining the boundary between NE and ER membranes and discuss the membrane protein network in *S. pombe*.

## Results

**Loss of Lem2 and Lnp1 disorders partitions of the NE and ER.** We have previously reported that the double disruption of *lem2⁺* and *lnp1⁺* (*lem2Δlnp1Δ*) induces severe synthetic growth defects in *S. pombe*, while the single disruption of each gene shows no growth defects[24]. To elucidate the functional relationship between

NE and ER proteins, we examined the growth of double mutants of Lnp1 with inner nuclear membrane proteins, Lem2 (HEH/LEM-domain protein), Man1 (another HEH/LEM-domain protein), or Bqt4 (Lem2-binding protein). We observed that, while *lem2Δlnp1Δ* induces severe growth defects in minimum medium (Edinburgh minimal medium with glutamate (EMMG)) and rich medium (Yeast Extract with Supplements (YES)), *man1Δlnp1Δ* or *bqt4Δlnp1Δ* grew normally in either medium (Supplementary Fig. 1). These results suggest that Lnp1 has a unique genetic interaction with Lem2 among these inner nuclear membrane proteins.

We then investigated the cellular phenotypes associated with severe growth defects in *lem2Δlnp1Δ* cells. We first observed the NE morphology by expressing mCherry-tagged Ish1 (Ish1-mCherry) as an NE marker[32,33] in wild-type (WT) and mutant cells (Fig. 1a). Remarkably, >85% of the *lem2Δlnp1Δ* cells displayed an abnormal NE morphology, a phenotype that was rarely (0.3–3.9%) observed in WT, *lem2Δ*, and *lnp1Δ* cells (Fig. 1a, b). Abnormal nuclear shapes deviating from the circular shape, disordered membranes extending from the nucleus (Fig. 1a, arrows), and highly aggregated membrane structures near the nucleus and in the cytoplasm (Fig. 1a, arrowheads) were observed in *lem2Δlnp1Δ* cells. We also examined localization of the nuclear pore complex (NPC) using green fluorescent protein (GFP) fusion of various nucleoporins (Nups), Cut11 (one of the transmembrane Nups), Nup97 (a component of the inner ring subcomplex), Nup62 (a component of the central channel), and Nup120 (a component of the outer ring subcomplex)[34]. These Nups in the mutant cells, carrying double disruption of *lem2⁺* and *lnp1⁺*, were localized at the nuclear periphery as in WT cells (Fig. 1c and Supplementary Fig. 2) but did not localize to the abnormally extended membranes outside the nucleus (Fig. 1c, arrows). These results suggest that the NE and ER membranes are disorganized by the double disruption of *lem2⁺* and *lnp1⁺*, leading to production of extra NPC-free membranes extending from the nucleus. To further examine whether the NE and ER are properly partitioned in the mutant cells, we observed the NE and ER membranes by expressing Ish1-mCherry and GFP-ADEL (an ER retention signal)[35] as a marker protein, respectively (Fig. 1d). Ish1-mCherry was localized at the NE and GFP-ADEL was localized at the perinuclear and cortical ERs in WT, *lem2Δ*, and *lnp1Δ* cells (Fig. 1d, WT, *lem2Δ*, and *lnp1Δ*), whereas in *lem2Δlnp1Δ* cells, Ish1-mCherry and GFP-ADEL were localized at the NE, the deformed ER, and the highly aggregated membranes (Fig. 1d, *lem2Δlnp1Δ*); quantification of localization of Ish1-mCherry and GFP-ADEL showed that their separate localization at perinuclear and cortical ER was lost in *lem2Δlnp1Δ* cells (Supplementary Fig. 3). In addition, Rtn1 protein, which participates in tubular ER network formation and is localized in the cortical ER[36], was also mislocalized in the aggregated membranes but not in the perinuclear ER in *lem2Δlnp1Δ* cells (Fig. 1e, arrow). These results suggest that the boundary between the NE and ER membranes is lost in *lem2Δlnp1Δ* cells. Because the double disruption mutant of *lem2⁺* and *lnp1⁺*, but not single disruption mutants, causes severe deformations of the NE or ER membranes, Lem2 and Lnp1 seem to have redundant functions in partitioning the NE and ER membrane compartments.

**Lem2 and Lnp1 are necessary for the nuclear barrier function.** To examine whether the severe deformations of the NE and ER membranes in *lem2Δlnp1Δ* cells are associated with breakage of the nuclear barrier function, we observed localization of the nuclear proteins by expressing nuclear localization signal (NLS) tagged with GFP-GST (GFP-GST-NLS). We assessed the nuclear barrier function by quantifying fold enrichment of the

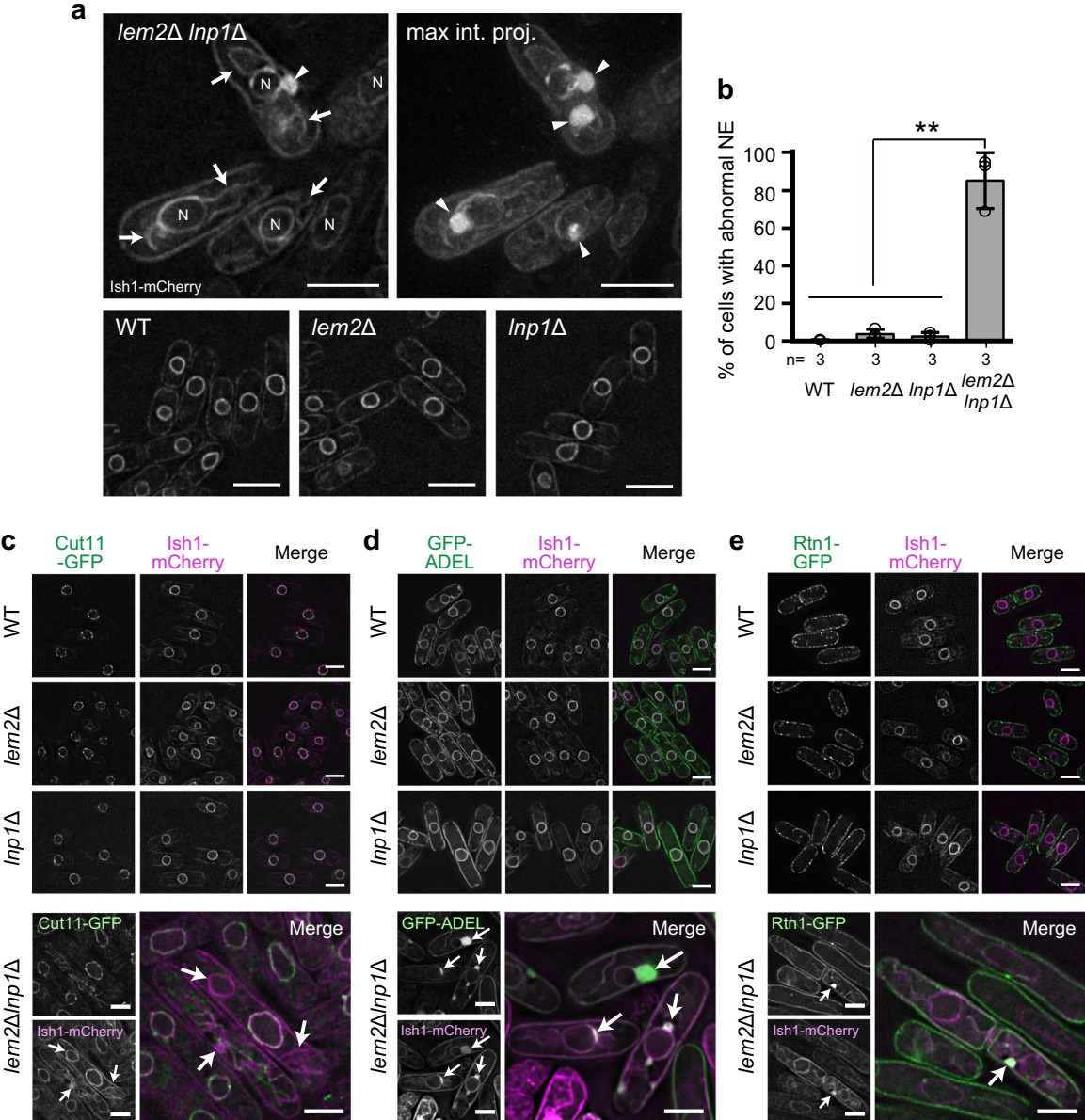

**Fig. 1 Severe NE and ER membrane defects observed in *lem2Δlnp1Δ* cells. a** NE structure of *lem2Δlnp1Δ* cells. The NE was labeled with Ish1-mCherry. A single-section image (upper left) and a maximum-intensity projected image (upper right) of *lem2Δlnp1Δ* cells are shown; single-section images are shown for wild-type (WT), *lem2Δ*, and *lnp1Δ* cells in the lower panels. N represents the nucleus. Arrows and arrowheads indicate disordered and highly aggregated membrane structures, respectively. Scale bar represents 5 μm. **b** Quantification of NE defects. Percentages of cells with abnormal NE were calculated from three independent experiments and plotted as mean ± standard deviation. *n* indicates the number of experiments. Open circles represent the percentage of individual experiment. *p* Values are from Tukey's test. **p < 0.01. **c** Effects on NPC structure. Cut11-GFP was co-expressed with Ish1-mCherry in the WT, *lem2Δ*, *lnp1Δ*, and *lem2Δlnp1Δ* cells, respectively. Arrows indicate an abnormal nuclear membrane. Scale bar represents 5 μm. **d** Effects on ER structure. GFP-tagged ER retention signal (GFP-ADEL) was co-expressed with Ish1-mCherry in the WT, *lem2Δ*, *lnp1Δ*, and *lem2Δlnp1Δ* cells, respectively. Arrows indicate an abnormal nuclear membrane. Scale bar represents 5 μm. **e** Effects on Rtn1 localization. Rtn1-GFP was co-expressed with Ish1-mCherry in the WT, *lem2Δ*, *lnp1Δ*, and *lem2Δlnp1Δ* cells, respectively. Arrow shows an abnormal nuclear membrane. Scale bar represents 5 μm.

GFP signal in the nucleus (nucleus/cytoplasm ratio) (Fig. 2a–e). The GFP signal was accumulated in the nucleus in WT, *lem2Δ*, and *lnp1Δ* cells (Fig. 2a–c), while it severely leaked from the nucleus to the cytoplasm in *lem2Δlnp1Δ* cells (arrows in Fig. 2d). Leakage of the GFP signal was observed during both interphase and mitosis. The percentage of cells with leakage during mitosis was 97% (143/147) in *lem2Δlnp1Δ* cells, while it was 0.46% (1/216) in WT, 1.1% (2/182) in *lem2Δ*, and 3.0% (7/233) in *lnp1Δ* cells. To dissect the timing of the mitotic leakage, we co-expressed Atb2-mCherry to visualize the mitotic spindle simultaneously with GFP-GST-NLS (Fig. 2f and

Supplementary Fig. 4). In *lem2Δlnp1Δ* cells undergoing mitosis, leakage started immediately after mitotic spindle formation, reached a maximum during chromosome segregation, and continued until mitotic spindle disassembly (Fig. 2f; see Fig. 2g, h for quantification), suggesting that the mitotic leakage correlates with SPB insertion/extrusion. In contrast, only transient and faint leakage during chromosome segregation was observed in the WT and single mutants (Supplementary Fig. 4). During interphase in *lem2Δlnp1Δ* cells, the transient leakage of GFP-GST-NLS occasionally occurred (time of leakage is indicated by red arrows in Fig. 2f, g). Such transient leakage

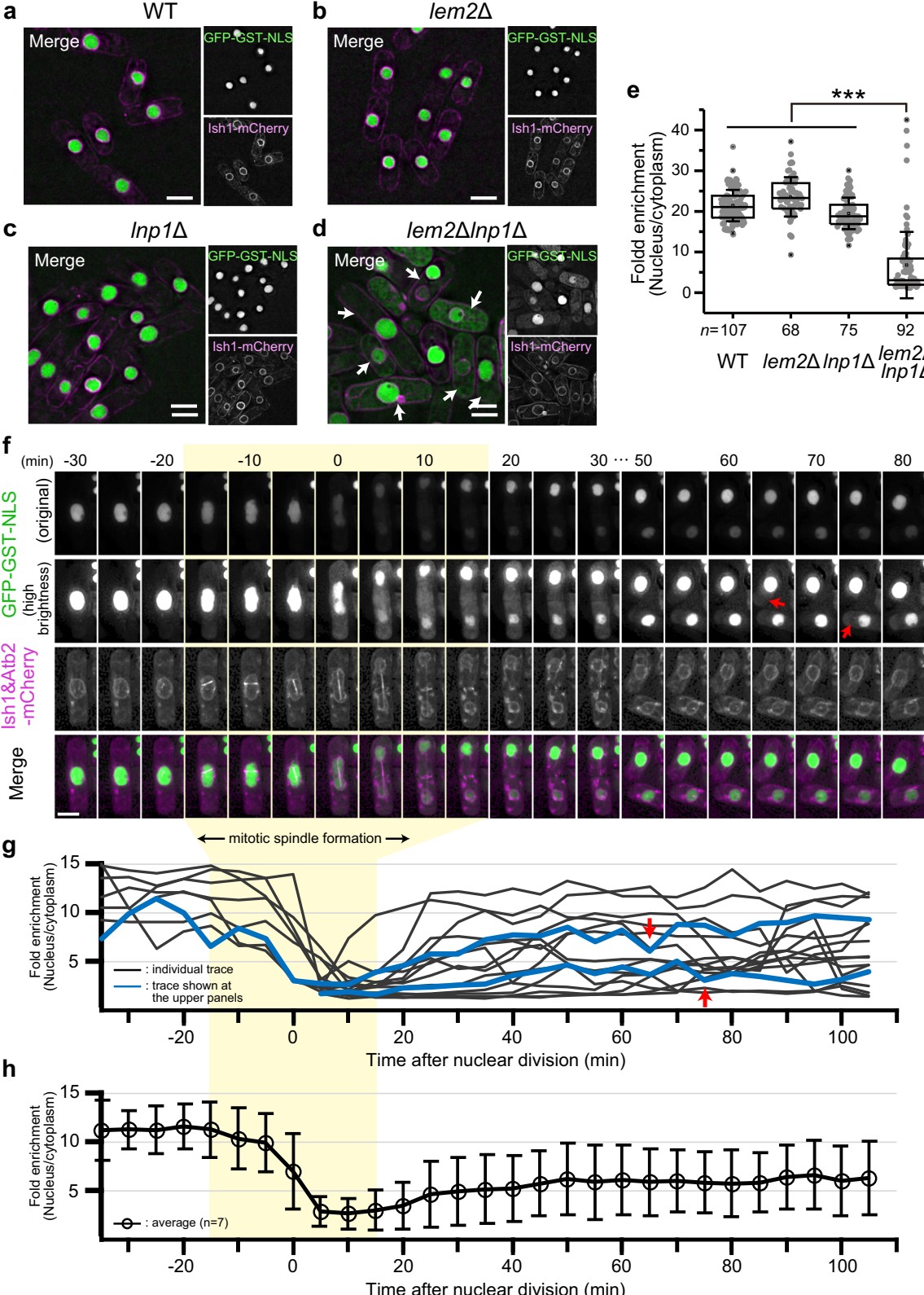

repeatedly occurred at random timings in each individual cell (see individual traces in Fig. 2g; Supplementary Movie 1) but not observed in WT, *lem2Δ*, and *lnp1Δ* cells (Supplementary Fig. 4, Supplementary Movies 2–4). As shown in Fig. 2h, variations of the fold enrichment values were larger in *lem2Δlnp1Δ* cells than in WT, *lem2Δ*, and *lnp1Δ* cells (compare with Supplementary Fig. 4), reflecting the transient nuclear leakage that occurred at random timings in *lem2Δlnp1Δ* cells. Because the leakage of the GFP signal was transient and the signal was able to recover in the nucleus during a relatively short period of time, it is unlikely that nuclear transport is defective in *lem2Δlnp1Δ* cells. Taken together, these observations suggest that the nuclear barrier function is lost in the absence of Lem2 and Lnp1.

**Fig. 2 Nuclear protein leakage occurs in *lem2Δlnp1Δ* cells. a–d** The *lem2Δlnp1Δ* cells showed a leaky phenotype of GFP-GST-NLS. GFP-GST-NLS (green) and Ish1-mCherry (magenta) were co-expressed in WT (**a**), *lem2Δ* (**b**), *lnp1Δ* (**c**), and *lem2Δlnp1Δ* (**d**) cells. Arrows indicate cells showing leakage of GFP-GST-NLS signal. Scale bar represents 5 μm. **e** Quantification of nuclear enrichment of GFP-GST-NLS signal. Fold enrichment of GFP-GST-NLS signal in the nucleus (nucleus/cytoplasm) was quantified (gray dots) by using unprocessed original images and plotted as a box-and-whisker plot. The horizontal lines in the box indicate the upper quartile, median, and lower quartile, from top to bottom, respectively; the whisker indicates standard deviation. *n* indicates the total cell number counted. *p* Values are from Steel–Dwass test. ***$p < 0.001$. **f–h** Leakage of nuclear proteins occurred independent of the cell cycle. **f** The *lem2Δlnp1Δ* cells expressing GFP-GST-NLS, Ish1-mCherry, and Atb2-mCherry were observed in living state during vegetative cell growth. Single-section images after denoising and deconvolution are shown as a time course. High-brightness images for GFP-GST-NLS are shown to present the leakage. Red arrows indicate the timing of leakage at the interphase. Scale bar represents 5 μm. **g, h** Fold enrichment of GFP-GST-NLS in the nucleus was quantified over time. The values from seven individual mitotic events were plotted (**g**). The blue lines represent the quantification results displayed in **f**. Mean ± standard deviation of the seven quantifications is plotted in **h**. Red arrows indicate the timing of leakage at the interphase.

**Lem2 and Lnp1 maintain the boundary between the NE and ER.** Loss of NE and ER partitioning and nuclear barrier function implied that NE and ER membranes were largely disordered in *lem2Δlnp1Δ* cells. To examine the membrane structures in *lem2Δlnp1Δ* cells that show nuclear protein leakage, we observed those cells by correlative light and electron microscopy (CLEM; Fig. 3). In the cells with nuclear protein leakage, abnormally developed membrane structures were observed: (1) highly aggregated membranes associated with the NE (Fig. 3a, b; white arrows), (2) nuclear membrane rupture (Fig. 3a–c; green arrows), (3) multi-layered nuclear membrane (Fig. 3c; purple arrows), (4) multi-layered membrane vesicles distinct from the nucleus (Fig. 3b, c; blue arrows), and (5) membrane invagination (Fig. 3b–d; yellow arrows). In addition, vacuole-like structures were often observed inside the nucleus (25/49 cells, Fig. 3a, c; arrowheads). Serial section images of the nucleus sometimes showed a crystal-like structure penetrating the nuclear membranes and fused with the vacuolar structures (Fig. 3a, d; double arrowheads). Such abnormal membrane structures were not observed in WT cells (0/89 cells, Fig. 3e, f). These results show that the boundary between the NE and ER is lost in *lem2Δlnp1Δ* cells and thereby suggests that Lem2 and Lnp1 play a role in maintaining the nuclear boundary.

**Domains of Lem2 and Lnp1 for their cooperative functions.** To understand the functional relationship between Lem2 and Lnp1, we examined the molecular domains required for cell growth. To do this, protein fragments of Lem2 were expressed in *lem2Δlnp1Δ* cells and the growth recovery was determined by a spot assay (Fig. 4a). Lem2 fragments lacking the HEH/LEM-domain (ΔLEM) or the bqt4-binding domain (Δ200-307) fully recovered growth to the level of the full-length Lem2 (Fig. 4a, +ΔLEM and +Δ200-307; compare with +Lem2), indicating that these two domains are dispensable for cell viability in the absence of Lnp1. Lem2 fragments lacking the LuC domain or the N-terminal domain (N and LuC, respectively) partially recovered growth (Fig. 4a, +N and +LuC), whereas the fragment lacking the C-terminal domain (NLu) fully recovered the growth (Fig. 4a, compare +N with +NLu). These results suggest that the N-terminal and luminal domains, but not the C-terminal domain, are necessary for the Lnp1-involving Lem2 function. To further investigate the role of the luminal domain, a mutant in which the luminal region of Lem2 was substituted by that of Man1 (Lem2N-Man1Lu-Lem2C; Supplementary Fig. 5a) were expressed in *lem2Δlnp1Δ* cells and the growth recovery was determined by a spot assay. A fragment of Lem2N-Man1Lu-Lem2C fused with GFP (Lem2N-Man1Lu-Lem2C-GFP) was localized to the NE (Supplementary Fig. 5); however, it displayed a very weak growth recovery (Fig. 4a, +Lem2N-Man1Lu-Lem2C). This result suggests that the luminal region of Lem2 plays an important role in cell viability in the absence of Lnp1.

Similarly, to identify functional domains in Lnp1, we examined the growth recovery effect of different Lnp1 fragments in *lem2Δlnp1Δ* cells (Fig. 4b). Lnp1 lacking the N-terminal 37 amino acid residues (ΔN) displayed a weak growth recovery (Fig. 4b, +ΔN). Since human Lunapark is myristoylated at the second glycine residue[37], which is conserved in *S. pombe* Lnp1, we examined the effect of an un-myristoylated point mutant (Lnp1-G2A). Expression of the Lnp1-G2A fragment in *lem2Δlnp1Δ* cells recovered the growth defect (Fig. 4b, +G2A). This result suggests that myristoylation has a minor effect on the N-terminal function of Lnp1. A more striking effect was observed with Lnp1 fragments lacking the lunapark domain, which is a highly conserved domain from yeasts to humans. Expression of the fragments lacking lunapark domains (Fig. 4b, +Δ104-334, +Δ177-334, and +Δ177-231) in *lem2Δlnp1Δ* cells conferred negligible growth recovery. In contrast, the fragment bearing the lunapark domain (Δ232-334) fully recovered the growth to the level of the full-length Lnp1 (Fig. 4b, +Δ232-334). These results indicate that the N-terminal and lunapark domains of Lnp1 play a crucial role in cell viability in the absence of Lem2.

We then examined the intracellular localization of these Lnp1 fragments by expressing them as a GFP fusion protein. The full-length Lnp1 was localized to both the cortical and perinuclear ER (Fig. 4c, Lnp1). The N-terminal deletion (ΔN) and the un-myristoylated Lnp1-G2A (G2A) showed the same localization as that of the full-length Lnp1 (Fig. 4c, ΔN and G2A). Lnp1 fragments lacking the lunapark domain dispersed through vacuoles as confirmed by co-staining with vacuole membrane-staining reagent FM4-64[38] (Fig. 4c, Δ104-334, Δ177-334, and Δ177-231, and Supplementary Fig. 6). In contrast, the Lnp1 fragment bearing the lunapark domain (Δ232-334) recovered its localization to the cortical and perinuclear ER (Fig. 4c). These results indicate that the conserved lunapark domain is crucial for proper localization of Lnp1.

Collectively, these domain analyses indicate that the luminal region of Lem2 and the cytoplasmic N-terminal and lunapark domains of Lnp1 are necessary for their genetic interaction although other factors may be involved in connecting Lem2 and Lnp1.

**Apq12 compensates Lnp1 functions in the absence of Lem2 and Lnp1.** To further understand the molecular basis of how Lem2 and Lnp1 cooperatively participate in partitioning between NE and ER membranes, we attempted to identify genes that genetically interact with the *lem2+* or *lnp1+* gene. For this purpose, we screened a multi-copy suppressor in the *lem2Δ lnp1-ΔN* genetic background because the Lnp1-ΔN fragment (ΔN) showed growth defects without changing its localization (see Fig. 4b, c). Thus we expected factors that compensated functions of Lem2 and Lnp1 to localize in the NE and ER membranes. In this screening, we transformed the *lem2Δ lnp1-ΔN* cells with a plasmid library of the *S. pombe* genome and obtained 223 well-grown

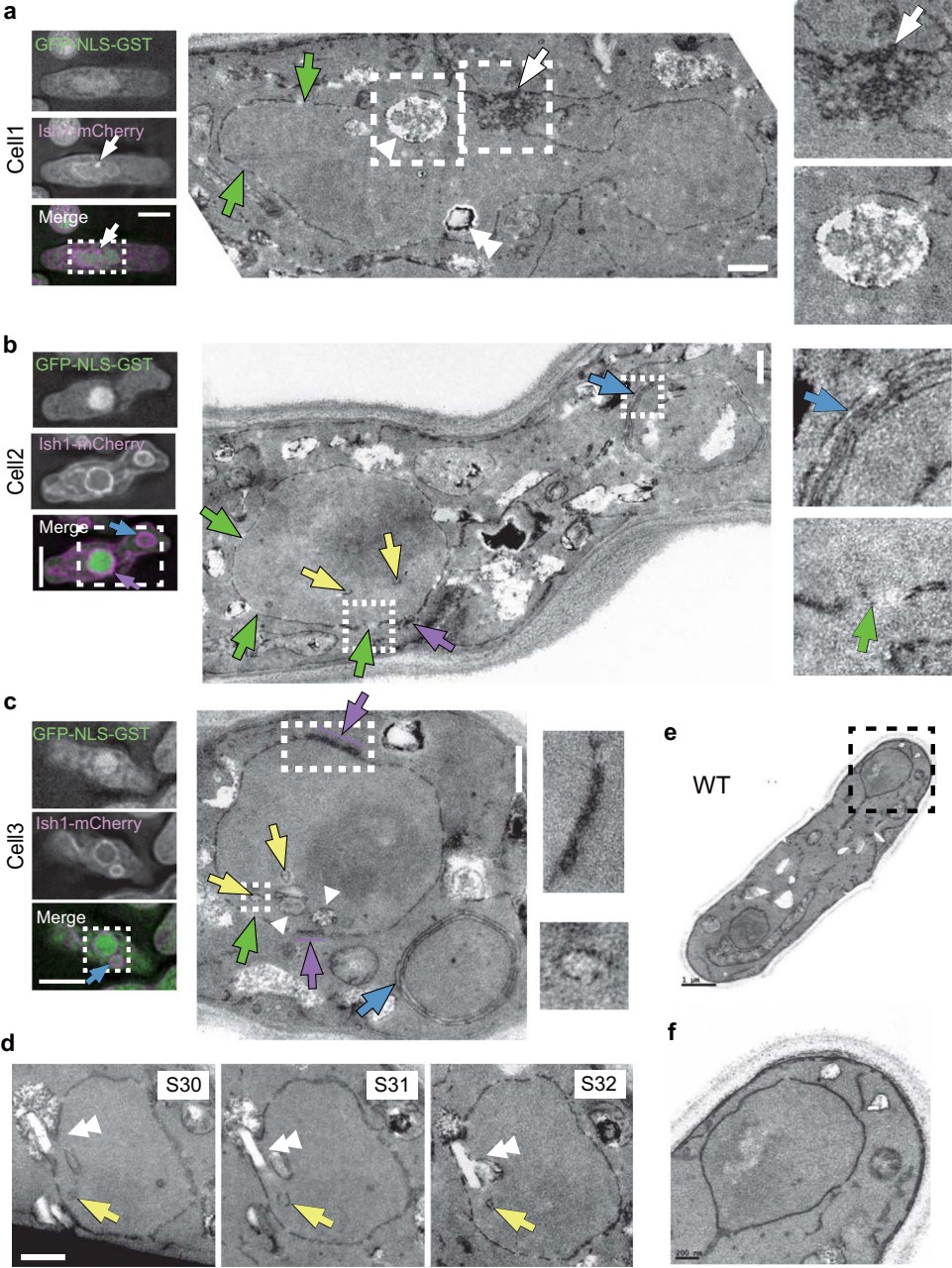

**Fig. 3 Abnormal NE and ER membranes in *lem2Δlnp1Δ* cells.** The *lem2Δlnp1Δ* cells expressing GFP-GST-NLS and Ish1-mCherry were observed by CLEM. **a–c** Single-section images after denoising and deconvolution from three independent cells are shown. The dashed square regions in the fluorescence images (left) correspond to the electron micrographs (middle). The dashed square regions in the electron micrographs are enlarged on the right. Serial sections of the nucleus in **c** are shown in **d**. Colored arrows indicate highly aggregated Ish1-positive membrane (white), nuclear membrane rupture (green), invaginated membrane (yellow), multi-layer nuclear membrane (purple), and multi-layer membrane vesicle distinct from the nucleus (blue). Arrowheads indicate a vacuole-like structure, and double arrowheads indicate a crystal-like structure penetrating the nuclear membrane. Scale bars in fluorescence and electron microscopy images represent 5 μm and 500 nm, respectively. **e**, **f** Electron micrographs of a WT cell as a control. The dashed square region in **e** is enlarged in **f**. Scale bars in **e** and **f** represent 1 μm and 200 nm, respectively.

suppressor colonies among approximately 60,000 transformants. PCR analysis or DNA sequencing revealed that 186 suppressor colonies carried the *lem2+* gene, 29 carried the *lnp1+* gene, and 8 carried other genes (see "Methods"). We determined the DNA sequences of the 8 unidentified suppressors and identified *apq12+* in 4 suppressors, *sec8+* in 1 suppressor, and *vid27+* in 2 suppressors as a responsible gene for suppression. These three genes rescued the growth defect of *lem2Δ lnp1-ΔN* cells to varying extents (Fig. 5a). Among them, *apq12+* encoding an ER membrane protein Apq12 almost fully rescued the growth defect to the

level comparable with that of *lem2+* and *lnp1+*. The rescue by *apq12+* overexpression on a multi-copy plasmid was also observed in *lem2Δlnp1Δ* cells (Supplementary Fig. 7a), strongly suggesting that Apq12 can compensate for the function of either Lem2 or Lnp1.

We next examined whether Apq12 rescued the NE and ER membrane defects. Expression of Apq12 under the constitutive *adh1* or *adh11* promoter on the chromosome rescued the growth to a level similar to that of the multi-copy plasmid (Supplementary Fig. 8). Thus we used the chromosomal expression from the

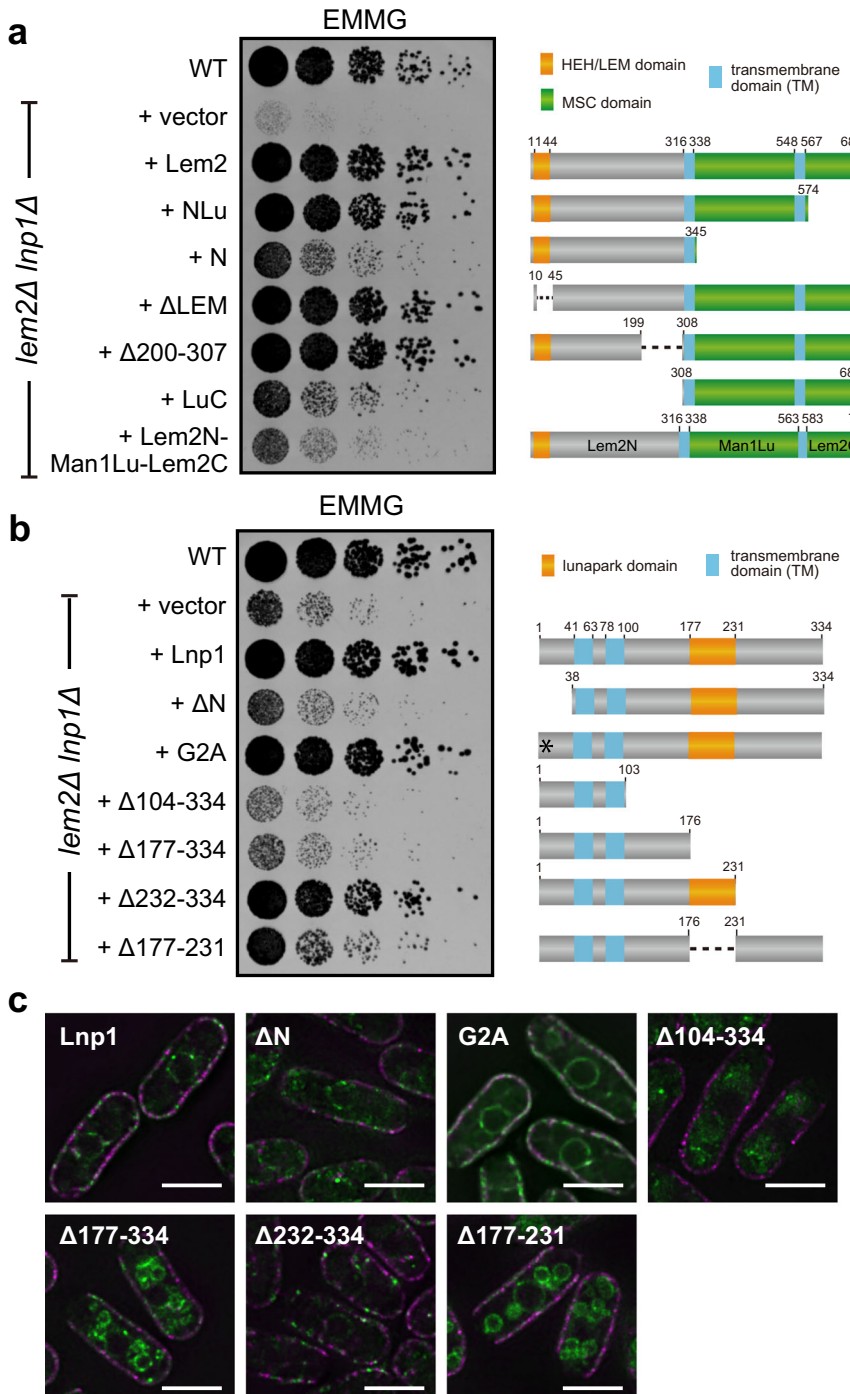

**Fig. 4 Functional domains of Lem2 and Lnp1. a** Lem2 fragments indicated on the left were expressed in *lem2Δlnp1Δ* cells. Diagram of these fragments is shown on the right. Fivefold serially diluted cells were spotted on EMMG plates, and growth of these cells was observed after 3 days. **b** Lnp1 fragments indicated on the left were expressed in *lem2Δlnp1Δ* cells. Diagram of these fragments is shown on the right. The asterisk indicates the mutation site of G2A. Fivefold serially diluted cells were spotted on EMMG plates, and growth of these cells was observed after 3 days. **c** Localization of Lnp1 fragments GFP-tagged Lnp1 fragments (green) and cortical ER membrane marker Rtn1-mRFP (magenta) were expressed in wild-type cells. Scale bar represents 5 μm.

*adh1* or *adh11* promoter in the following experiments to stabilize the expression levels. Under these conditions, expression of Apq12 rescued the NE and ER membrane defects during interphase, that is, abnormal NE shape (Fig. 5b, c) and nuclear protein leakage (Fig. 5d, e). However, the defect in nuclear protein leakage during mitosis was not restored (arrows in Fig. 5d; Supplementary Fig. 9, and Supplementary Movie 5).

To reveal the genetic relationship among Apq12, Lem2, and Lnp1, we constructed double deletion mutants of the *lem2+*, *lnp1+*, and *apq12+* genes and examined the growth and the NE morphology of these mutant cells (Fig. 5f–h). No growth defects were observed in cells of the single deletion of *apq12+* (*apq12Δ*), as in *lem2Δ* and *lnp1Δ* (Fig. 5f). Growth defects and abnormal NE shape were observed in *lem2Δapq12Δ* similar to *lem2Δlnp1Δ* but not in *lnp1Δapq12Δ* (Fig. 5f–h). These results suggest that Apq12 has an overlapping function with Lnp1, both of which are necessary in the absence of Lem2, and that the overlapping functions of Lnp1 and Apq12 are redundant with Lem2

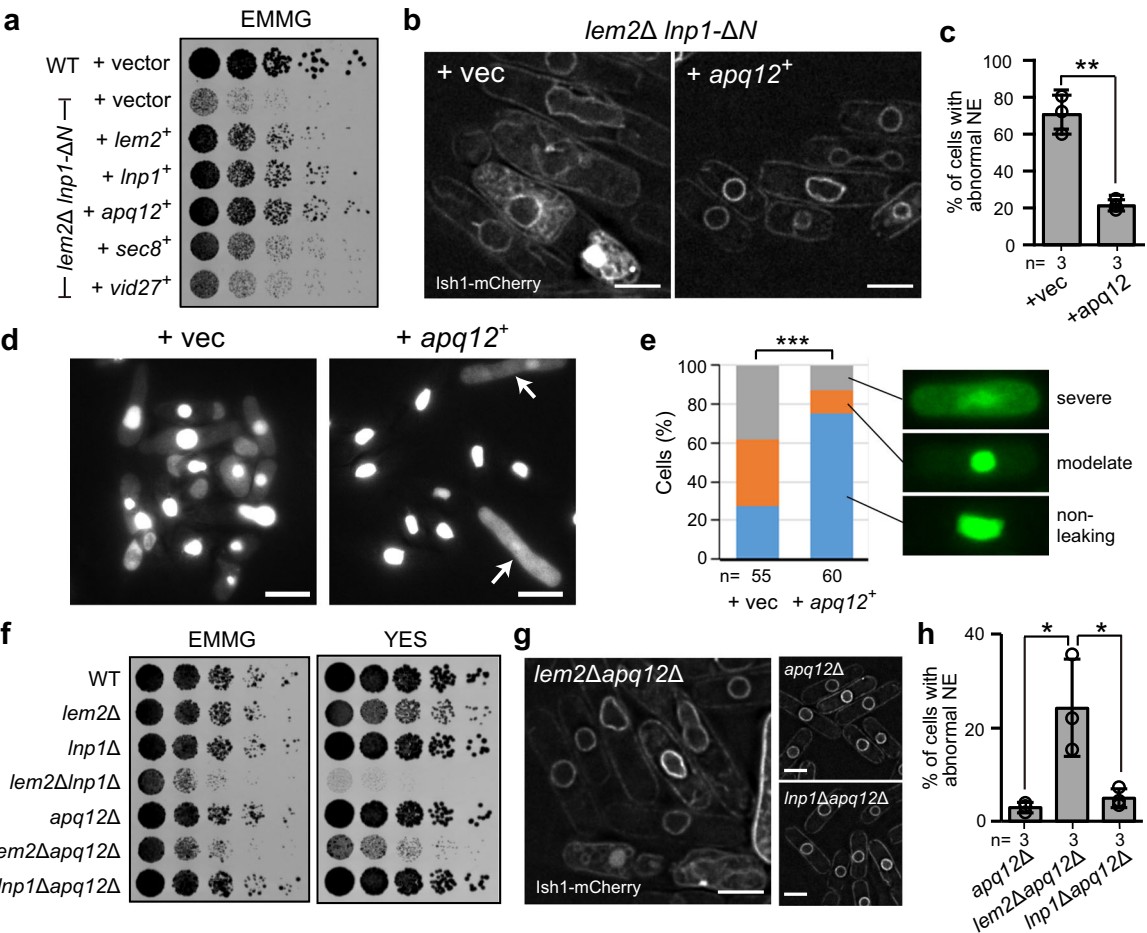

**Fig. 5 Apq12 rescues the nuclear membrane defect in *lem2Δlnp1Δ* cells by bypassing Lnp1 function. a** Genes indicated on the left were expressed using the multi-copy plasmid in the *lem2Δ lnp1-ΔN* cells. Fivefold serially diluted cells were spotted on EMMG plates, and growth of these cells was observed after 4 days. **b** Cells of *lem2Δ lnp1-ΔN* added back with *apq12+* (+*apq12+*) and control vector (+vec) were labeled by Ish1-mCherry to observe the NE structure. Scale bar represents 5 μm. **c** Percentages of the cells with abnormal NE are plotted as mean ± standard deviation. *n* indicates the number of experiments. Open circles represent the percentage of individual experiment. *p* Value is from unpaired two-tailed Student's *t* test. \*\**p* < 0.01 (*p* = 0.0014). **d** GFP-GST-NLS was expressed in *lem2Δ lnp1-ΔN* cells added back with *apq12+* (+*apq12+*) and a control vector (+vec) to observe nuclear protein leakage. Arrows indicate mitotic cells. Scale bar represents 10 μm. **e** Fold enrichment of GFP-GST-NLS signal in the nucleus (nucleus/cytoplasm) observed in **d** was quantified and classified into three levels according to its leakage: non-leaking (ratio ≥10), moderate (10> ratio ≥3), and severe (ratio <3). Percentages of the cells are plotted as a cumulative bar chart. *n* indicates the total cell number counted. Asterisks denote significant differences; \*\*\**p* < 0.001 by χ² test (*p* = 2.1 × 10⁻⁶). **f** Growth of the strains indicated. Fivefold serially diluted cells were spotted on the EMMG or YES plate, and growth of these cells was observed after 3 and 2 days, respectively. **g** The NE of the indicated mutant cells was labeled with Ish1-mCherry. Scale bar represents 5 μm. **h** Percentages of the cells with abnormal NE are plotted as mean ± standard deviation. *n* indicates the number of experiments. Open circles represent the percentage of individual experiment. *p* Value is from Tukey's test. \**p* < 0.05.

functions in maintaining the membrane boundary between the NE and ER.

**Vps4 is mislocalized in the absence of Lem2 and Lnp1**. Overexpression of Apq12 recovered the NE–ER boundary in the *lem2Δ lnp1Δ* genetic background as described above. Considering that *APQ12* genetically interacts with *CHM7*, which encodes an ESCRT-III component in *S. cerevisiae*[4,11,39], the ESCRT-III complex may be involved in maintenance of the Lem2-Lnp1-dependent NE-ER boundary. To test this hypothesis, we examined whether overexpression of ESCRT-III-related *vps4+* and *cmp7+* genes could rescue the phenotypes of *lem2Δ lnp1-ΔN* cells. Since *BRR6* has been identified as a genetic interactor of *APQ12* in *S. cerevisiae*[40,41], we also tested the overexpression of *brr6+* in *lem2Δ lnp1-ΔN* cells. Our results showed that overexpression of *vps4+*, but not *cmp7+* or *brr6+*, partially restored the growth defect in *lem2Δ lnp1-ΔN* cells (Fig. 6a). We confirmed that

overexpression of *vps4+* also restored the growth of *lem2Δlnp1Δ* cells (Supplementary Fig. 7c). We also tested whether overexpression of these genes had any effects on cell growth and found no effects on the growth of WT and *lem2Δ* cells (Supplementary Fig. 7b). These results suggest that higher levels of Vps4 is necessary to restore the growth in the absence of Lem2 and Lnp1. As a possible cause for this, we speculated that functional endogenous Vps4 may be reduced in the absence of Lem2 and Lnp1.

To test this possibility, we expressed Vps4 tagged with mCherry (Vps4-mCherry) in WT and mutant cells and examined their protein levels (Fig. 6b). First, we confirmed that Vps4-mCherry was functional by performing the following experiments: (1) substitution of endogenous Vps4 with Vps4-mCherry rescued the growth defect of *vps4Δ* cells; (2) overexpression of Vps4-mCherry rescued the growth defect of *lem2Δ lnp1Δ* cells (Fig. 6b; Supplementary Fig. 7c). Using these strains, we

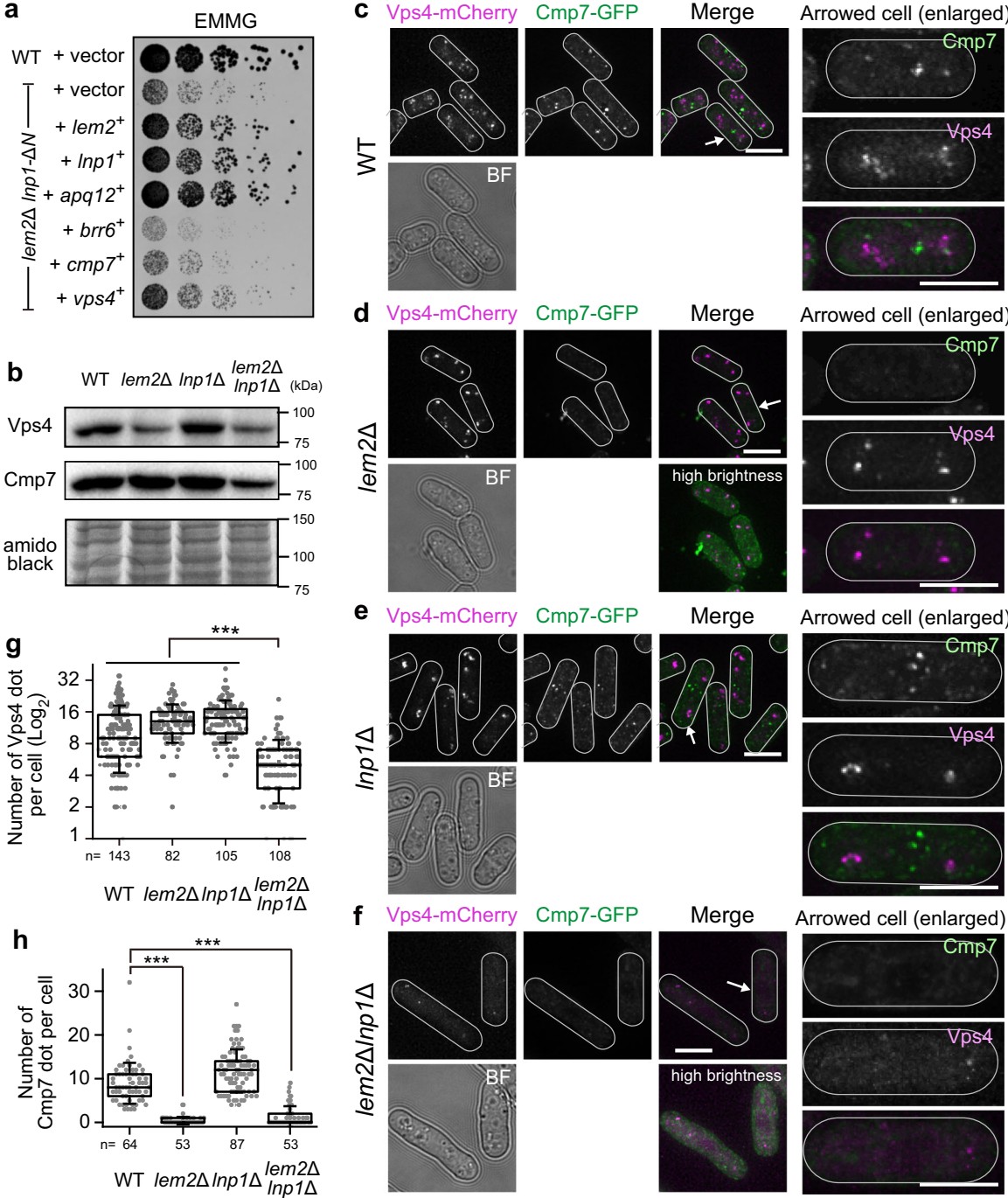

**Fig. 6 Lem2 and Lnp1 cooperatively affect localization of Vps4. a** Genes indicated on the left were expressed from the multi-copy plasmid in the *lem2Δ lnp1-ΔN* cells. Fivefold serially diluted cells were spotted on the EMMG plate, and growth of these cells was observed after 4 days. **b** Protein levels of Vps4 and Cmp7. The WT, *lem2Δ*, *lnp1Δ*, and *lem2Δlnp1Δ* cells expressing Vps4-mCherry and Cmp7-GFP were subjected to SDS-PAGE. Vps4-mCherry was detected using an anti-RFP antibody, which recognizes mCherry; Cmp7-GFP was detected using an anti-GFP antibody. As a loading control, the total protein amount was detected by amidoblack staining (amidoblack). Molecular weight marker is indicated on the right. The full, uncropped blot images are shown in Supplementary Fig. 14. **c–f** Localization of Vps4 and Cmp7. Vps4-mCherry (magenta) and Cmp7-GFP (green) were expressed in WT (**c**), *lem2Δ* (**d**), *lnp1Δ* (**e**), and *lem2Δlnp1Δ* (**f**) cells. Projected images after deconvolution are shown. White lines indicate the outline of the cell. Arrowed cells are enlarged and shown on the right. Scale bar represents 5 µm. High-brightness images of Vps4 and Cmp7 for *lem2Δ* and *lem2Δlnp1Δ* cells are shown at the right bottom. BF represents bright field image. **g**, **h** Quantification of the number of Vps4 (**g**) and Cmp7 (**h**) dots. The number of dots per cell was counted and plotted as a box-and-whisker plot: horizontal lines in the box indicate upper quartile, median, and lower quartile, from top to bottom, respectively; the whisker indicates standard deviation. *n* indicates the total number of cells counted. *p* Values are from Steel–Dwass test. ***$p < 0.001$.

compared protein levels of WT with those of mutant cells by western blotting. Results obtained showed that the protein level of Vps4 decreased in *lem2Δ* and *lem2Δlnp1Δ* cells compared with that in WT and *lnp1Δ* cells (Fig. 6b), suggesting that Lem2 affected the stability of Vps4. We next examined the intracellular localization of Vps4-mCherry in these strains. In WT cells, the fluorescence signal of Vps4-mCherry was localized at the cytoplasmic dots (Fig. 6), and this localization overlapped with the localization of ESCRT-III components, Did2, Did4, Vps24, Vps32, and Vps60 (*S. pombe* homologs of CHMP1–CHMP5, respectively; Supplementary Fig. 10a), except for Cmp7; Cmp7-GFP formed cytoplasmic dots distinct from the Vps4 dots (Fig. 6c). Bright dots of Vps4-mCherry were localized adjacent to the FM4-64-positive vacuolar membranes (Supplementary Fig. 10b), suggesting that some fractions of Vps4 are associated with endosomes, as described previously[42,43]. In *lnp1Δ* cells, Vps4-mCherry and Cmp7-GFP showed the cytoplasmic dots distinct from each other (Fig. 6e), same as in WT cells. In *lem2Δ* cells, Vps4-mCherry remained at the cytoplasmic dots, whereas Cmp7-GFP lost the dotted localization and diffused to the cytoplasm (Fig. 6d), suggesting that Lem2 may affect Cmp7-dependent ESCRT-III functions. In *lem2Δlnp1Δ* cells, both Vps4-mCherry and Cmp7-GFP lost their dotted localization and were diffused to the cytoplasm (Fig. 6f). Accordingly, the number of Vps4 dots remained unchanged in WT, *lem2Δ*, and *lnp1Δ* cells but significantly decreased in *lem2Δlnp1Δ* cells (Fig. 6g). The number of Cmp7 dots significantly decreased in *lem2Δ* and *lem2Δlnp1Δ* cells (Fig. 6h). Other ESCRT-III components (Did2, Did4, Vps24, Vps32, and Vps60) remained at the cytoplasmic dots in *lem2Δ*, *lnp1Δ*, and *lem2Δlnp1Δ* cells (Supplementary Fig. 11). These results suggest that Lem2 and Lnp1 are necessary for the localization of Vps4 at the endosome. Taken together, Lem2 and Lnp1 synergistically affect the stability and localization of Vps4.

It has been previously reported that depletion of Vps4 induces a strong deformation in the membrane phenotype and growth defect, and these phenotypes are rescued by disruption of the *cmp7*[+] gene[5]. Because *vps4Δ* cells exhibit phenotypes similar to those observed in *lem2Δlnp1Δ* cells, we speculated that disruption of the *cmp7*[+] gene may bypass the phenotypes observed in *lem2Δlnp1Δ* cells by analogy with *vps4Δ* cells. To test this possibility, we disrupted the *cmp7*[+] gene with *lem2*[+] and *lnp1*[+]. However, *cmp7*[+] gene disruption did not rescue the growth defect or the disordered membrane phenotype in *lem2Δlnp1Δ* cells (Supplementary Fig. 12, *lem2Δlnp1Δcmp7Δ*). Next, we tested whether the nuclear barrier function was defective in *cmp7Δ* cells. For this purpose, we observed nuclear localization of GFP-GST-NLS during the cell cycle (Supplementary Fig. 13 and Supplementary Movie 6). Results showed that no obvious nuclear protein leakage was observed in *cmp7Δ* cells throughout the cell cycle, unlike in *lem2Δlnp1Δ* cells (Supplementary Fig. 13; compare with Fig. 2h), although a slight leakage was observed at the end of mitosis (see 20 min after the nuclear division in Supplementary Fig. 13). Thus it is unlikely that the nuclear protein leakage in *lem2Δlnp1Δ* cells results from the dysfunction of Cmp7. Taken together, these results indicate that Cmp7 did not bypass the functions of Lem2 and Lnp1.

**Vps4 localization at the SPB during mitosis.** We next examined localization of Vps4 during mitosis because Vps4, together with Cmp7, is predicted to participate in repairing the NE hole at the site where the SPB is inserted and extruded. We observed GFP-tagged Vps4 co-expressed with an SPB marker Sfi1 tagged with mCherry and found that Vps4-GFP was localized to the SPB during mitosis in WT and *lnp1Δ* cells (Fig. 7a, WT and *lnp1Δ*).

However, the SPB localization of Vps4-GFP was lost in *lem2Δ* and *lem2Δlnp1Δ* cells (Fig. 7a, *lem2Δ* and *lem2Δlnp1Δ*), suggesting that localization of Vps4 at the SPB during mitosis depends on Lem2. Cmp7-GFP showed similar results to Vps4 during mitosis in these mutants (Fig. 7b). Considering that the localization of Vps4 was independent of Lem2 during interphase (see Fig. 6c–f), this result suggests that the mitotic localization of Vps4 is regulated differently from that of the interphase.

**Apq12 recovers Vps4 localization in the absence of Lem2 and Lnp1.** Because our results suggested that Apq12 has overlapping functions with Lnp1, we examined the effect of Apq12 over-expression on Vps4 localization. As shown in the previous section, dotted localization of Vps4-mCherry found in WT was lost in *lem2Δlnp1Δ* cells (Fig. 6f). Interestingly, overexpression of Apq12 in *lem2Δlnp1Δ* cells restored the dotted localization of Vps4 (Fig. 8a); the number of Vps4 dots that had decreased in *lem2Δlnp1Δ* cells was also recovered (Fig. 8b). Western blotting showed that overexpression of Apq12 restored the protein levels of Vps4, which was decreased in *lem2Δ* and *lem2Δlnp1Δ* cells to a comparable level of WT cells (Fig. 8d). Thus Apq12 is likely to stabilize Vps4 and maintain its proper localization during interphase. In contrast, during mitosis, Apq12 overexpression did not restore the SPB localization of Vps4 in *lem2Δlnp1Δ* cells (Fig. 8e). We also tested the effect of Apq12 overexpression on Cmp7 and found no effects on its localization and protein levels (Fig. 8a, c, d). These results suggest that Apq12 has overlapping functions with Lnp1 in Vps4 localization and stability.

**Genetic interaction of the NE/ER membrane proteins.** Our findings suggest that genetic interaction of membrane proteins maintains the membrane boundary between NE and ER, as depicted in Fig. 8f. Lem2 and Lnp1 have a redundant function required for proper Vps4 localization. Apq12 has an overlapping function with Lnp1. Because Vps4 is necessary for membrane remodeling[44,45], Lem2 and Lnp1/Apq12 cooperatively maintain the membrane boundary between NE and ER by remodeling membranes through the functions of Vps4 and the ESCRT-III complex independently of Cmp7.

## Discussion

In this report, we have demonstrated that the membrane boundary between the NE and ER is severely affected in the absence of Lem2 and Lnp1. In *lem2Δlnp1Δ* cells, various types of membrane defects, such as highly aggregated membrane continuous with the outer nuclear membrane, nuclear membrane invagination, and vacuole-like structure inside the nucleus, were observed (Fig. 3), strongly suggesting a role for Lem2 and Lnp1 in maintaining the boundary for proper partitioning of the NE and ER membrane compartments. This is consistent with a recent report which demonstrates that Lem2 and Lnp1 buffer the membrane flow into and out of the NE[46].

The phenotypes observed in *lem2Δlnp1Δ* cells are remarkably similar to those in *vps4Δ* cells. The deletion of *vps4*[+] gene causes a severely disordered nuclear membrane phenotype and strong growth defect in *S. pombe*[5]. It has also been reported that ESCRT-III filament and Vps4 cooperatively remodel the membrane and induce membrane scission by using ATPase activity of Vps4[44,45]. Our findings (Fig. 6a, e and Supplementary Fig. 11) consistently indicate that Lem2 and Lnp1 cooperatively regulate Vps4 localization. Thus it is likely that Vps4 dysfunctions in *lem2Δlnp1Δ* cells, and consequently *lem2Δlnp1Δ* cells exhibit phenotypes similar to *vps4Δ* cells, that is, abnormal membrane and growth defect. In light of a previous report showing that Lem2 and Lnp1 act as a barrier to membrane flow and contribute to the control of

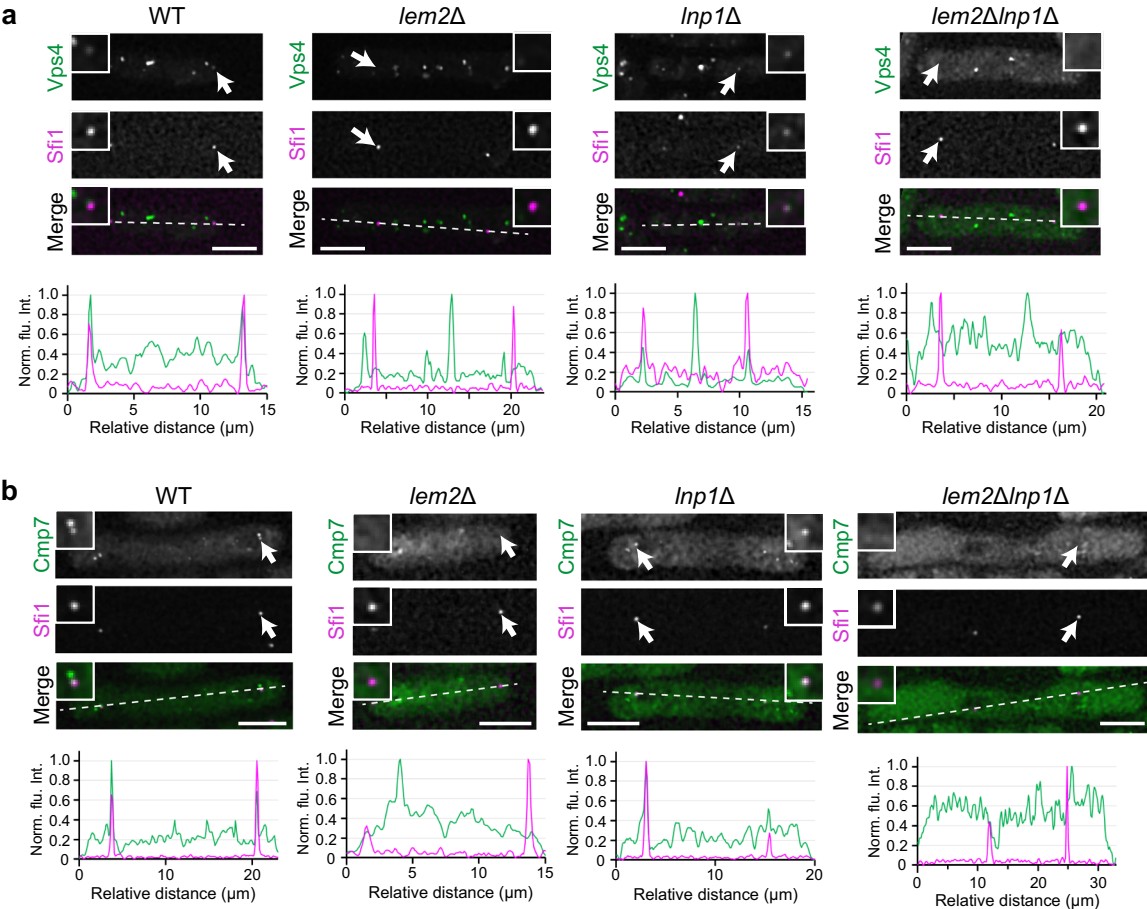

**Fig. 7 Vps4 and Cmp7 accumulate at the SPB in Lem2-dependent manner.** Vps4-GFP (**a**) and Cmp7-GFP (**b**) (green) were co-expressed with an SPB marker Sfi1-mCherry (magenta) in WT, *lem2Δ*, *lnp1Δ*, and *lem2Δlnp1Δ* cells as indicated and observed in live. Insets represent enlarged images of the arrowed SPB. Fluorescence intensities of Vps4, Cmp7, and Sfi1 along the dashed line are plotted after normalization using unprocessed original images. Scale bar represents 5 μm.

nuclear size[46], we speculate that membranes to be transported between organellar compartments including NE and ER are not cropped from organelles because of dysfunction of Vps4 in *lem2Δlnp1Δ* cells. It has been reported that ESCRT-III components have a Vps4-binding motif necessary to recruit Vps4 to the ESCRT-III complex[47–49]. However, our findings (Fig. 6g) showed that localization of Vps4 on endosomes was significantly decreased in the absence of Lem2 and Lnp1, even in the presence of ESCRT-III complex bearing the functional Vps4-binding motif. This implies that Lem2 and Lnp1 may provide a structural scaffold that ensures interaction of Vps4 with the ESCRT-III complex. Based on this idea, we propose that Lem2 and Lnp1 cooperatively maintain the NE–ER boundary by regulating the recruitment of Vps4 to the ESCRT-III complex.

Cmp7-dependent NE membrane sealing by ESCRT-III at the end of mitosis has been reported in *S. cerevisiae*, *Schizosaccharomyces japonicus*, *S. pombe*, and human cells[5,11,18,43] as well as in interphase[17,18]. Cmp7 is recruited to the reforming NE by Lem2 to form the ESCRT-III complex, and then Vps4 dissociates the ESCRT-III complex to complete the NE sealing[5,6,8]. In this study, we found that Lem2 was required for the localization of both Cmp7 and Vps4 at the mitotic SPB, only where NE sealing occurs in *S. pombe* (Fig. 7). Our findings are consistent with the previous findings described above. However, Cmp7 may not be a critical factor for sealing the interphase NE breakage that occurs in *S. pombe lem2Δlnp1Δ* cells because deletion of the *cmp7*⁺ gene caused no nuclear protein leakage or growth defects during

interphase (Supplementary Fig. 13; also Gu et al.[5]). Thus it is likely that NE sealing associated with SPB extrusion during mitosis is Cmp7 dependent, but the NE–ER boundary during interphase is regulated independently of Cmp7 in *S. pombe*.

It is reported that Lem2 and Chm7 play a role in NPC surveillance in *S. cerevisiae*[50]. However, Lem2 does not seem to play a role in *S. pombe*. Instead, *S. pombe* Lem2 play a role in the control of lipid metabolism and trafficking. All suppressors of *lem2Δlnp1Δ* identified in this study (Apq12, Sec8, and Vid27) have a function related to lipid metabolism. Apq12 participates in neutral lipid biosynthesis, especially for steryl esters and triacylglycerols[40,41,51] Sec8 and Vid27 are involved in membrane vesicle trafficking in many biological processes as a component of exocyst complex and vacuolar import and degradation family, respectively[52–54]. In addition, our recent study has demonstrated that the synthetic lethality of *lem2Δbqt4Δ* is rescued by overexpression of very-long-chain fatty acid elongase Elo2[23]. Therefore, Lem2 seems to play an important role in regulating membrane composition and/or structure. This study sheds light on the involvement of a membrane protein network in the maintenance of the nuclear membrane barrier.

## Methods

***S. pombe* strains and culture media**. All *S. pombe* strains used in this study are listed in Supplementary Data 1. The minimum medium used was EMMG or EMMG5S (EMMG with five supplements); the rich medium used was YES[55]. Cells were cultured in an appropriate medium at 30 °C for 3–6 days, depending on their

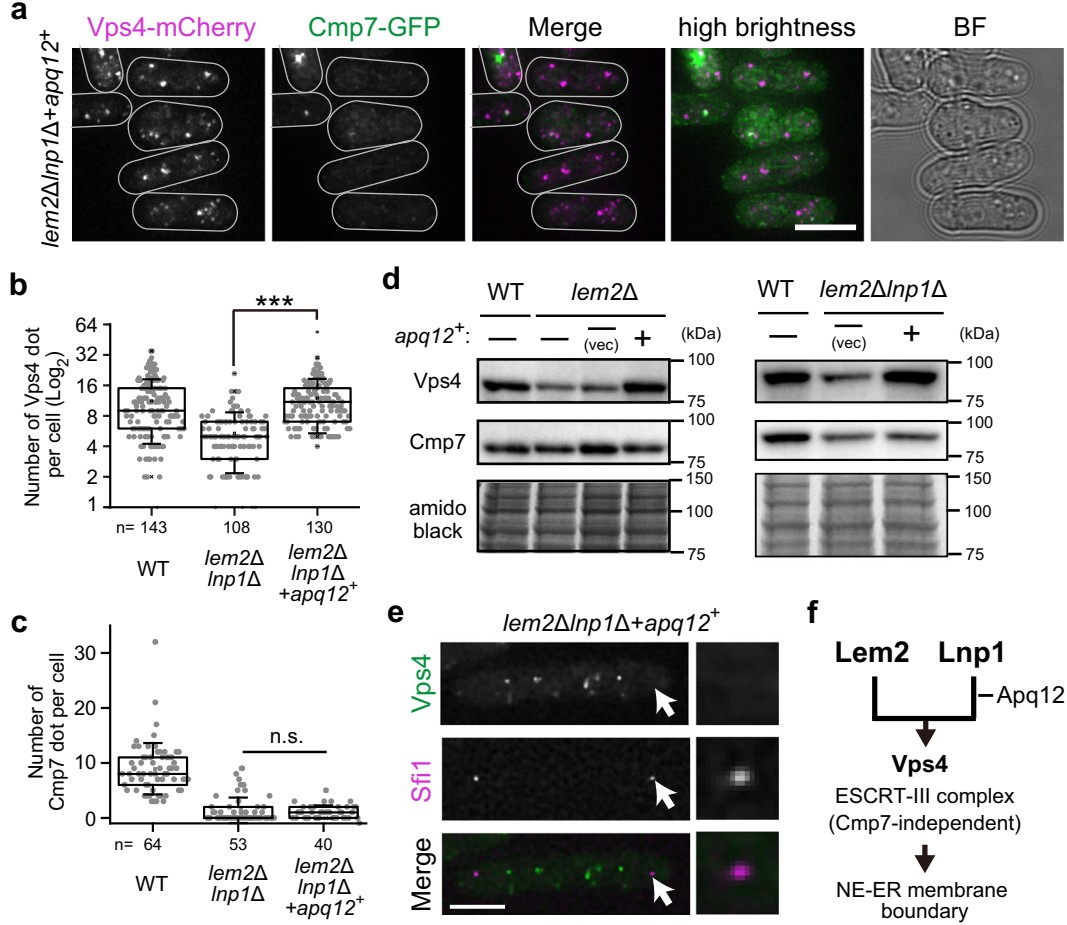

**Fig. 8 Overexpression of Apq12 restores the Vps4 localization in *lem2Δlnp1Δ* cells during interphase. a** Localization of Vps4 and Cmp7. Vps4-mCherry (magenta) and Cmp7-GFP (green) were expressed in Apq12-overexpressing *lem2Δlnp1Δ* cells. Projected images after deconvolution are shown. White lines indicate the outline of the cell. BF represents bright field image. Scale bar represents 5 μm. **b**, **c** Quantification of the number of Vps4 and Cmp7 dots. The number of Vps4 (**b**) and Cmp7 (**c**) dots per cell were counted and plotted as a box-and-whisker plot: horizontal lines in the box indicate the upper quartile, median, and lower quartile, from top to bottom, respectively; the whisker indicates standard deviation. The data for WT and *lem2Δlnp1Δ* cells are reproduced from Fig. 6g, h for comparison. *n* indicates the total cell number counted. *p* Values are from Steel–Dwass test. ***$p < 0.001$, n.s.: no significance ($p > 0.05$). **d** Protein levels of Vps4. The WT and Apq12-overexpressing *lem2Δ* (left panels) and *lem2Δlnp1Δ* cells (right panels) were subjected to SDS-PAGE. The protein levels of Vps4 and Cmp7 were detected as described in Fig. 6b. The full, uncropped blot images are shown in Supplementary Fig. 14. **e** Mitotic localization of Vps4 in Apq12-overexpressing *lem2Δlnp1Δ* cells. Vps4-GFP (green) was co-expressed with an SPB marker Sfi1-mCherry (magenta) in Apq12-overexpressing *lem2Δlnp1Δ* cells and observed in live. Insets on the right represent enlarged images of the arrowed SPB. Scale bar represents 5 μm. **f** Genetic interactions in maintaining the NE–ER boundary.

growth. The strains including *lem2Δ* were maintained in EMMG medium as previously reported[23,24,56]. For selection, 100 μg/ml G418 disulfate (Nacalai Tesque, Kyoto, Japan), 200 μg/ml hygromycin B (FUJIFILM Wako Pure Chemical Corp., Osaka, Japan), 100 μg/ml nourseothricin sulfate (WERNER BioAgents, Jena, Germany), and 100 μg/ml blasticidin S (FUJIFILM Wako Pure Chemical Corp.) were added to the media, respectively.

**Gene disruption, integration, and tagging**. Gene disruption, integration, and tagging were performed using the direct chromosome integration method, as previously described[57,58]. Briefly, ~500 bp upstream and downstream genomic regions from open reading frames (ORFs) of interest were amplified as the homologous recombination sites for both the 5′ and 3′ ends by PCR using KOD plus neo or KOD One (TaKaRa Bio Inc., Kusatsu, Japan). These PCR products are used as primers to amplify a cassette for disruption, integration, and tagging, including selection markers. The cassette was transformed to *S. pombe*, and the transformants were selected on an appropriate selection plate. The correct disruptions, integrations, and tagging were confirmed by genomic PCR using KOD Fx neo (TaKaRa Bio Inc.) from both the 5′ and 3′ ends.

**Plasmid construction**. All plasmids used in this study were constructed using the NEBuilder system (New England Biolabs, Ipswich, USA) according to the manufacturer's protocol. Restriction enzymes were purchased from TaKaRa Bio Inc. (Kusatsu, Japan) and New England Biolabs (Ipswich, USA). DNA sequences coding

Lem2 fragments were amplified by PCR from the genome and cloned into authentic *lem2* promoter-harboring pCST3-FLAGHA into *Bam*HI and *Bgl*II sites[56]. DNA sequences coding Lnp1 including the promoter (~700 bp upstream from the start ATG codon), the protein-coding region, and the terminator (~1000 bp downstream from the stop codon) were amplified by PCR using the cDNA library pTN-FC9 as a template (obtained from the National BioResource Project Yeast in Japan), respectively, and then cloned into pYC36[59] at the *Bam*HI site.

**Spot assay**. *S. pombe* cells were cultured overnight in EMMG medium at 30 °C to attain the logarithmic growth phase. Fivefold serially diluted cells were spotted on EMMG plates (initial cell numbers spotted are $4.0 \times 10^3$ cells) and cultured at 30 °C for 2–4 days.

**Multi-copy suppressor screening**. *S. pombe* genomic library, pTN-L1, was provided by the National BioResource Project Yeast in Japan. H1N532 cells (*h⁻ leu1-32 lem2Δ::kanʳ lnp1Δ::hph lys1⁺::lnp1-ΔN*) of $8.0 \times 10^7$ were transformed with 6.5 μg of the genomic library, plated on EMMG, and then incubated at 30 °C for 4–5 days. Approximately 60,000 transformants were screened on EMMG plates. The colonies displaying better growth were picked and re-streaked on the EMMG plate to check the growth recovery, and 223 suppressor colonies were obtained. Plasmids in the cells were isolated by Cameron's method[60]. Plasmids containing the *lem2⁺* and *lnp1⁺* genes were identified by PCR analysis. After removing the suppressors containing the *lem2⁺* and *lnp1⁺* genes (186 and 29 suppressors, respectively), we

determined the genomic region inside the plasmids from the remaining 8 suppressors by sequencing. Because most of the plasmids contained multiple genes, the genomic region of a single gene including ORFs, 5′ and 3′ untranslated regions (UTRs), and ~1.0 kb upstream and downstream sequences from the UTR were amplified again by PCR and inserted into the pAL-SK vector (obtained from the National BioResource Project Yeast in Japan) at the BamHI site. The plasmids were then introduced into the H1N532 cells to identify the gene responsible for rescuing growth.

**Microscopic observation.** *S. pombe* cells were observed using the DeltaVision Elite system (GE Healthcare Inc., Chicago, USA) equipped with cool SNAP HQ$^2$ (Photometrics, Tucson, USA) or pco.edge 4.2 sCMOS (PCO, Kelheim, Germany), ×60 PlanApo N OSC oil-immersion objective lens (numerical aperture [NA] = 1.4, Olympus, Tokyo, Japan) or the DeltaVision OMX (GE Healthcare Inc.) equipped with Cascade II (Photometrics) and ×100 UPlanSApo silicone-immersion objective lens (NA = 1.35, Olympus). Intracellular localization of GFP-S65T (designated GFP throughout this study), mRFP, and mCherry fusion proteins was observed in living cells. For live cell imaging, cells were cultured overnight in EMMG medium at 30 °C to attain the logarithmic growth phase before placing them onto the glass-bottom dish. The cells were attached to the glass via soybean lectin (Sigma-Aldrich, St. Louis, USA) and covered with EMMG medium. Chromatic aberration except time-lapse observation for multi-color images was corrected using the Chromagnon software (v0.70) using a bleed-through fluorescence image as a reference[61]. Optical section images were presented after deconvolution using the built-in SoftWoRx software unless otherwise noted. Denoising by the ND-safir program[62] was performed before deconvolution when necessary. The brightness of the images was changed using the Adobe Photoshop CS6 software for better visualization, without changing the gamma settings.

**CLEM imaging.** CLEM imaging was performed as described previously[63,64]. Briefly, *lem2Δ lnp1Δ* cells were cultured overnight in EMMG medium. Cells were attached onto a dish with a gridded coverslip (ibidi, Martinsried, Germany) as described above and fixed by replacing the medium with a fixative (2% glutaraldehyde in 0.1 M sodium phosphate buffer, pH 7.2). During fixation, the cells were observed by fluorescence microscopy (DeltaVision system) and then further fixed with fresh fixative at 4 °C for 2 h. After washing with buffer, the cells were postfixed in 1.2% KMnO$_4$ overnight, dehydrated, and embedded in epoxy resin. Serial sections of 80-nm thickness were stained with uranyl acetate and lead citrate and analyzed using a transmission electron microscope (JEM1400Plus, JEOL, Japan) at 80 kV.

**Western blotting.** Cells cultured in an appropriate minimum medium as described above were harvested at mid-log phase by centrifugation. The cells were lysed in 1.85 N NaOH for 15 min on ice. Proteins were precipitated by adding 27.5% trichloroacetic acid. After washing the precipitated proteins with ice-cold acetone twice, the proteins were resolved in 2× Laemmli sodium dodecyl sulfate (SDS) sample buffer without dye. Protein concentration was quantified by bicinchoninic acid assay (Thermo Fisher Scientific, Waltham, MA, USA; cat. #23225) according to the manufacturer's protocol. The same amount of total protein was subjected to SDS–polyacrylamide gel electrophoresis and then transferred to polyvinylidene difluoride membrane. The fluorescent proteins were probed with anti-GFP monoclonal (JL8, 1:2000 dilution; TaKaRa Bio Inc.) and anti-RFP polyclonal (PM005, 1:5000 dilution; MBL, Nagoya, Japan) antibodies, respectively, and detected using chemiluminescence (ImmunoStar LD, FUJIFILM Wako Pure Chemical Corp.). The membrane was stained with amidoblack to determine the amount of loaded protein.

**Statistics and reproducibility.** For the quantification of fluorescent signals, nuclear and cytoplasmic regions in the original unprocessed image were manually marked as regions of interest (ROIs) using the Fiji software[65]. After subtraction of the background noise, measured outside the cells, the mean fluorescent intensities of the ROIs were measured. Fold enrichment (mean intensity ratio of nucleus/cytoplasm) was calculated for each cell and plotted as a box chart. Significance was assessed by Steel–Dwass test for non-parametric multiple comparison by Microsoft Excel. The number of cells examined are shown in figure legends.

For phenotype quantification, images were collected from at least three independent experiments. Percentages of the phenotype were calculated in each experiment (total number of cells examined were >500), and the values are represented as mean with standard deviation. Significance was assessed by Tukey's test or unpaired two-tailed Student's *t* test using Microsoft Excel.

**Reporting summary.** Further information on research design is available in the Nature Research Reporting Summary linked to this article.

## Data availability
The data supporting the findings of this study are available within the paper and its Supplementary Information files. Further datasets are available from the corresponding authors upon reasonable request.

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

## Acknowledgements

We thank Ms. Chizuru Ohtsuki for her technical assistance and Dr. Haruhiko Asakawa and Dr. Takeshi Sakuno for insightful discussion. We also acknowledge the National BioResource Project Yeast in Japan for providing the *S. pombe* genomic DNA library and plasmids. This study was supported by JSPS KAKENHI Grant Numbers JP19K06489 (to Y. Hirano), JP19K23725 (to Y. Kinugasa), JP25116006, JP17H03636 and JP18H05528 (to T.H.), and JP17H01444, JP18H05533, and JP19K22389 (to Y. Hiraoka). This work was also supported by a grant from Brain Mapping by Integrated Neurotechnologies for Disease Studies (Brain/MINDS) by AMED under the Grant Number JP18dm0207002 (to T.S. and S.S.), Dr. Yoshifumi Jigami Memorial Fund from The Society of Yeast Scientists, and Kiriyama Foundation (to Y. Hirano).

## Author contributions

Y. Hirano, T.H., and Y. Hiraoka conceived and designed the experiments and acquired funding. Y. Hirano, T.H., Y. Kinugasa, and Y. Kubota performed and analyzed the majority of the experiments. H.O., T.S., S.S., and T.H. performed and analyzed electron microscopic experiments. Y. Hirano generated figures with input from Y. Kinugasa, H.O., T.S., S.S., T. H., and Y. Hiraoka. Y. Hirano, T.H., and Y. Hiraoka wrote the manuscript with input from all the co-authors.

## Competing interests

The authors declare no competing interests.
