## [Peer Review File · Communications Biology]

Reviewers' comments:

Reviewer #1 (Remarks to the Author):

Hirano et al. present an interesting manuscript documenting roles for the INM protein Lem2 and the ER-morphometric protein Lnp1 in regulating membrane homeostasis at the nuclear envelope and controlling the biology of the ESCRT-III complex through Vps4. In recent years, there has been much interest in the biology of ESCRT in membrane sealing at the nuclear envelope where it plays a role during mitotic exit through the coordination of the ESCRT-III component Chmp7 via Lem2. Interestingly, the mitotic defect in membrane dealing is a delay in compartmentalisation, suggesting the existence of a compensatory mechanism. Additionally, ESCRT plays a role in nuclear envelope homeostasis (and NPC quality control in yeast) during interphase, but whether this pathway operates through the cell division pathway is unknown. For these reasons, the manuscript presented is interesting, as it tries to separate interphase and mitotic effects of the Lem2/Cmp7 axis. However, I found the manuscript a too observational and both mechanistically underdeveloped and over-interpreted toward the end, with some strange structuring whereby some figures were only mentioned in the discussion. The manuscript is also very reliant upon interpretation of function from the viability of multiply-deleted and rescued yeast strains, which (given the plethora of functions ascribed to ESCRT proteins, Lem proteins and Lnp1) makes data interpretation confusing.

The 1st half of the manuscript, in which the authors build upon their 2016 publication was generally convincing; the excessive membrane expansion and compartmentalisation defect in the *lem2Δ lnp1Δ* is clear, although how these two proteins (one regulating 3-way junction formation in the peripheral ER and the other regulating INM biology) work together to suppress these problems is unclear. Control movies would be helpful.

The defects in at the ultrastructural level are generally presented clearly, however, methods should be provided for the EM, rather than referencing their 2019 JCS publication, which in turn references their 2010 publication. What are all the white (unstained) regions – are these fixation artefacts? I'm surprised too at the presence of endosomes/vacuoles within the nucleus. As serial sections were taken, can you exclude that these are cytoplasmic regions that are enveloped by the nucleus? Are the 'crystal structure' that you refer to really just a fixation artefact? I think someone with more experience of EM than I ought to look at these.

The genetic interactions between Lem2's luminal domain and Lnp1 are clear from the data presented in Fig4, but unclear from a topological perspective. In discussion of this figure, it may be prudent to note that the Cmp7 interacting region of Lem2 (its MSC domain) is dispensable for rescuing the growth defect induced by *lnp1* loss.

When rescuing the growth defect in *lnp1Δ, lem2Δ*, the authors examined localisation of the rescue construct, however, it wasn't clear was whether membrane expansion in the *lnp1Δ, lem2Δ* was the cause of the viability defect? Does Lnp1 d177-231 (4C) rescue the growth defect in 4B? Also, are the Lnp1 fragments that localise to 'vacuole' structures really vacuoles, or are they extensions of the ER? An overexpression screen to rescue the growth defect in *lem2Δ lnp1Δ* identified Aqp14, previously identified by the Lusk/Kolling as genetically interacting with Cmp7. Restoration of membrane proliferation and NE permeability in Aqp14+ was clear and the authors conclude that Aqp14 functions in the Lnp1, but not the Lem2 pathway (L238/9).

Further viability assays concern the ability of overexpressed Vps4 and Cmp7 to rescue the growth defect in a *lem2Δ, lnp1-N* strain, which displays a weak growth defect. Vps4 was able to do this, but I think the conclusion that this means that Vps4 functions were downregulated in the absence of *lem2*

and Inp1 is an overinterpretation. Does Vps4 overexpression rescue a Lem2d Inp1d?

The authors find that Vps4-mCh, but not Cmp7-GFP, is destabilised in lem2d and lem2d Inp1d cells. Is endogenous Vps4 similarly affected? This is an interesting result, but I think the authors need to demonstrate why Vps4 is destabilised in the absence of Lem2. The authors quantify the number of Vps4+ endosomes in their cells, and find no difference in the absence of Lem2. Is there a difference in the cytosolic pool? Loss of Lem2 and Inp1 also destabilises Vps4, but also destabilises Cmp7 now, and reduces the endosomal pool of Vps4. I find this figure largely observational and the mechanism/biology underlying these observations is unclear. Currently, the destabilisation of Vps4 (Lem2-dependent) and the reduction in Vps4+ endosomes (Lem2 and Inp1 dependent) are not linked. Why is the endosomal pool of Vps4 lost in the double deletion? How are these observations related to the growth defect and membrane expansion (which appeared independent of the Lem2-Cmp7 axis; 4A)? Is the number of endosomes reduced in lem2d Inp1d, or just the number of Vps4+ endosomes and is this related to viability/proliferation? I'm afraid I find the observations in this figure unlinked and confusing.

The authors suggest that loss of Vps4 may hyperactivate ESCRT-III (L273), but I don't think that this interpretation is consistent with our understanding of how ESCRT-III functions; loss of Vps4 is likely to suppress ESCRT-III activity. The triple depletion of lem2, Inp1 and cmp7 is even more toxic than lem2d Inp1d, (which should be noted in the text) but the membrane proliferation phenotype is not as strong. This makes me question how related the membrane proliferation and viability phenotypes are. I don't think the fact that Cmp7 deletion failed to rescue the growth of lem2d, Inp1d is evidence that Cmp7 functions independently of Lem2 or Inp1 (L280, L281).

The authors next turn to the rescue of viability and membrane hyperproliferation in the Aqp12 overexpressing lem2d Inp1d yeast and find that this recovers both the endosomal pool of Vps4 and the expression levels of Vps4, but not those of Cmp7. How this pool is recovered is unclear. Is the destabilisation of Vps4 in lem2d similarly recovered by Aqp12 overexpression?

The data in Figure S8 is not discussed in the results section, but relates to Lem2-dependent SPB localisation of Cmp7 and Vps4 that is not rescued by Aqp12 overexpression.

I'm afraid I found the ESCRT-part of the manuscript confusing, observational and unlinked and fail to extract a clear message from the data. I could see no data in the manuscript supporting the conclusion (in the title) that Lem2 and Inp1 act to load Vps4 onto ESCRT-III. The discussion brings this 'loading' conclusion in, but this is at odds with the mechanism by which Vps4 is recruited by ESCRT-III proteins; all ESCRT-III members have MIMs to recruit Vps4 directly and any role for Lem2 or Inp1 in 'loading' ESCRT-III with Vps4 is confusing.

The discussion is very overinterpreted/speculative, especially from L362 to the end.

N numbers of independent experiments should be provided for all panels.

Reviewer #2 (Remarks to the Author):

Hirano et al focus on describing the role of three nuclear associated proteins lem2, Inp1, and apq12 in regulating nuclear envelope resealing. While lem2 and apq12 have been previously implicated in an ESCRT-dependent nuclear envelope resealing process, the authors suggest that there is a separate nuclear sealing/repair/maintenance mechanism that is Cmp7-independent. While this study provides

very little mechanistic insight into ESCRT-mediated resealing, the Cmp7-independent mechanism is intriguing. This study would need to go much further to substantiate their model (Figure 9 right side), however, or the authors must remove any mechanistic model prior to publication consideration.

Major points:

- 1) The Cmp7-independent model in Figure 9 is not plausible. How would ESCRT-III components and Vps4 access the ER-lumen? These proteins are synthesized in the cytosol and do not possess secretion signals that would allow for co-translational folding into the ER lumen and hence access the intramembrane nuclear space. This model must be removed or revised. I suggest that simply the bottom text be included since there is genetic evidence for a Cmp7-independent pathway.
- 2) There is no microscopic evidence that other ESCRT-III components are found on the surface of the nuclear envelope. The authors appear to loosely connect Vps4 foci to the other ESCRT-III components, however, there is no evidence that the colocalization between ESCRT-III and Vps4 foci (shown in SFig. 7) are residing on the nuclear envelope. These foci could represent the well characterized endosomal/multi-vesicular body functions of ESCRT-III and Vps4. Additionally, there is no evidence in Figure 6 that ESCRT-III foci are enriched on the nuclear membrane in the Cmp7 null background as is suggested by the mechanism in Figure 9. Further, do ESCRT-III foci become enriched at the nuclear envelope in the *lem2/lnp1* null background? The statement on lines 251-253 are incorrect given the data.
- 3) Lines 124-126 and Fig. 1e: The localization of Rtn1 is shown only in *lem2Δlnp1Δ* cells, with no WT control. To support the assertion that "tubular ER formation seemed to be disorganized in this mutant", the localization of Rtn1 in WT cells should be shown for comparison, so that the reader can see how its organization in the mutant cells is abnormal. WT control images are also missing for Rna1 localization (Line 142-144, Fig 2g). These comparisons would be useful for a general cell biology audience.
- 4) The frequency of Cmp7-GFP foci in Figure 6 and Figure 8 have not been quantified as they were for Vps4.
- 5) Figure 2f is not apparently replicated (bottom graph), can this be reproduced with similar magnitudes for fold GFP enrichment to assess error and/or biological variability?

Minor points:

- 1) The authors need to justify the statement 279-280 because it seems to be at odds with Gu, M. et al PNAS 2017. Knock-out of *lem2* alone prevents CHMP7 localization.
- 2) The authors use a number of ESCRT-III fluorescent protein fusions including Cmp7-GFP. While it appears that the *S. Pombe* field is using these ESCRT-III fluorescent protein fusions routinely, this reviewer cannot find any validation study of these probes. Is the mere survival of these organisms carrying fluorescent protein fusions sufficient evidence to prove that ESCRT-mediated functions are not impaired by the probe? This should at least be discussed.
- 3) SPB was not defined in the manuscript.
- 4) Line 77, the word "rapture" is used to describe the nuclear envelope. I believe the authors meant to use the word "rupture". This is also in Fig 9.
- 5) Lines 137-139 and Fig. 2f: N-values or some measure of the uncertainties should be included for the percentages of WT and single mutant cells exhibiting GFP-GST-NLS leakage.
- 6) Fig. 6a, It's not clear why *brr6+* was included in the experiment, or what to make of the data indicating that it further inhibited growth of the mutant cells rather than rescuing. Brr6 was not discussed in the text except that some of the titles on the reference list suggest that Brr6 forms a complex with Apq12, but I don't think the reader should have to look to the reference list to get a clue as to why this protein might be relevant.

Reviewer #3 (Remarks to the Author):

Lem2 and Lnp1 maintain the nuclear envelope-ER boundary by loading Vps4 to the ESCRT-III complex

In this manuscript Hirano et al, characterize the phenotypes associated to deletion of the nuclear membrane LEM domain containing protein Lem2 and the ER protein Lnp1. They find that both mutants show aberrant NPC-less membrane aggregations and expansions with mixing of ER and NE membranes. This phenotype is exacerbated in the double mutant *lem2Δ lnp1Δ* and is accompanied by loss of nuclear compartmentalization that affects cell viability. Based on this the authors suggest that both proteins function in maintaining the ER/NE membrane compartmentalization.

This conclusion is supported with nice CLEM images of these mutants where overdeveloped membrane structures and NE breakage are observed. They further identify Lnp1 and Lem2 functional domains by analysing the extent of suppression of these aberrant membrane phenotypes of the *lem2Δ lnp1Δ* mutant by different Lnp1 and Lem2 protein fragments.

They found that the ER protein *apq12* and *vps4*, a component of the ESCRT-III membrane remodelling system, are multi-copy suppressor of the growth and membrane phenotypes of the absence of Lem2 and Lnp1 function, and *apq12* in addition also recovers nuclear compartmentalization during interphase but not during mitosis. These results and also the strong genetic interaction between *lem2Δ apq12Δ* and *lnp1Δ lem2Δ* but not in *lnp1Δ apq12Δ* support the hypothesis that that *apq12* and Lnp1 function in the same pathway at the ER/NE different from the Lem2 pathway.

They further explore the relationship between these membrane regulators and the conserved ESCRT-III/Vps4 system that is required for proper membrane sealing and reformation and that has been poorly characterized in the fission yeast. They find that ESCRT-III is indeed regulated by two partially redundant pathways. Lem2 and Lnp1/*Apq12* are required for proper localization of Vps4 to endomembranes, whereas exclusively Lem2 is required for proper localization of the ESCRT-III adaptor Cmp7 to the NE at the SPBs during mitosis. This Cmp7 dependent function of ESCRT-III seems to be especially important during mitosis, where it is required to avoid nuclear leakage at time of SPB extrusion from the NE

The results presented here are convincing and the message of this study is interesting as it highlights important differences in the spatial regulation of a conserved membrane remodelling system. The distinction between ESCRT-III regulation at the NE and at other compartments of the endomembrane system, is of great interest in the field and the role of NE and ER proteins that regulate membrane structure and lipid composition in these processes reinforces a new and interesting role of different lipid domains in a variety of nuclear functions as has been demonstrated by the same authors in previous studies (Kinugasa et al, 2019). So I am in favour of its publication but I still have some points that might be clarified:

1. Introduction, lines 54-62.

It is inferred here that the ESCRT-III system coupled with Spastin is a conserved mechanism that operates to seal any membrane hole in *S.cerevisiae*, *S.pombe* and metazoans. To my knowledge this system (at least Spastin function) has only been described in human cells and not in yeasts and only in sealing those remaining holes that remain in contact with spindle MTs. I think this point might be easily clarified.

2. Figure 1e. There is only one panel showing Rtn1 localization in the double mutant. The authors should include a wt as a control as the wt localization of Rtn1 has not been previously shown in the manuscript (in fact in the figure legend it is included the wt strain).

3. Page 6 line 131-132. “with severe NE and ER membrane defects resulting in breakage of the nuclear sealing and nuclear transport defect.”

There is no data to this point that supports a defect in nuclear transport, at least in NPC-dependent transport, indeed NPCs localize properly. (See below, point 6)

4. Figure 2d. The mitotic leakage is very penetrant (97% of mitotic events). I wonder whether the loss of compartmentalization shown in the double mutant in fig 2 d in interphase, is the result of a defect in NE remodelling that occurs during mitosis (as shown in f) likely during SPB extrusion from the NE or whether there are other events during interphase such as SPB insertion that contribute to this phenotype. Can the authors show the frequency of interphase leakage? Can the authors find a correlation between the position and/or timing of SPB insertion and nuclear leakage? The fact that overexpression of Apq12 rescues the membrane defects of the lem2lnp1 mutant during interphase but not during mitosis is striking and suggest that indeed different mechanism regulate NE maintenance during interphase and during mitosis. Additionally, different components of the membrane system might be preferentially remodelled during interphase or during mitosis.

-

5. Figure 2 g / page 7, lines 142-143.

This experiment is confusing to me. If I am not mistaken, in a previous study, the same authors showed that Rna1 is cytoplasmic but once it enters the nucleus (by forcing its localization during interphase) it leads to a sort of “virtual nuclear envelope break down” (Asakawa et al, 2010) and the release of nuclear components. According to this, Rna1 entrance in the nucleus would promote the further release of nuclear components. I think that the GFP-GST-NLS or RFP-NLS are good markers to follow nuclear compartmentalization and that the result on Rna1 might distract from the important points of this study. In addition this experiment is not explained nor discussed in the main text so I suggest removing this experiment.

6. page 7 lines 156-158. Here again authors suggest that the loss of nuclear barrier function in the double mutant is caused by nuclear membrane rupture rather than transport defects or the NPC, but they do not discuss the reasons for this conclusion. The localization of GFP-GST-NLS (Figure 2F), and RFP-NLS (Figure 2g) in the nucleus and the reimport of these proteins once nuclear compartmentalization is recovered in the double mutant prove that NPCs are functional. I would make this point more explicit (maybe moving the conclusion in lines 156-158 to line 143).

7. page 12 lines 279-280. “therefore, Cmp7 seems to function through a pathway independent of Lem2 or Lnp1”. I think that this conclusion is not supported by the data showing that Cmp7 localization is strongly dependent on Lem2. Have the authors looked specifically whether deletion of Cmp7 affects the mitotic loss of compartmentalization? I understand from the results that the aberrant global nuclear membrane phenotypes might be independent on the specific loss of compartmentalization that occurs during mitosis (during SPB externalization for example) as this is not rescued by overexpression of Apq12 (Fig 5d).

8. Finally, the authors propose that these functions of Lem2, Lnp1 and Apq12 might be achieved through changes in membrane structure and/or lipid composition. In this sense, the same authors have shown in a previous study that the overexpression of the fatty acid elongase Elo2 rescues most of Lem2 described phenotypes and also the more penetrant NE aberrant phenotypes and loss of compartmentalization during mitosis of the double mutant lem2bqt4. I wonder whether overexpression of Elo2 would rescue the nuclear compartmentalization of lem2lnp1D specifically during mitosis (and perhaps Cmp7 localization)

Minor comments:

-Title: ER should not be abbreviated.

-Page 4 line 77: "rapture" should be "rupture".

- I am concerned about the size of some of the figures, especially when they are printed. As examples, fig 1c and d left panels. Figure S2 a, b and c left panels, and figure S8.

Response to Reviewers

We greatly appreciate the valuable comments from the reviewers and made changes to the manuscript according to those comments. Especially, we removed mechanistic models in former Figure 9 and related statements. Accordingly, we changed the title.

New title: **Lem2 and Lnp1 maintain the membrane boundary between the nuclear envelope and endoplasmic reticulum.**

Detailed point-by-point responses to the comments are described below. *Blue italic sentences are reviewer's comments*, black roman sentences are our responses, and **red font parts are quotation from the text** in this response letter. The response letter is followed by the manuscript with **the revised parts marked in red fonts**.

Reviewer #1

Hirano et al. present an interesting manuscript documenting roles for the INM protein Lem2 and the ER-morphometric protein Lnp1 in regulating membrane homeostasis at the nuclear envelope and controlling the biology of the ESCRT-III complex through Vps4. In recent years, there has been much interest in the biology of ESCRT in membrane sealing at the nuclear envelope where it plays a role during mitotic exit through the coordination of the ESCRT-III component Chmp7 via Lem2. Interestingly, the mitotic defect in membrane sealing is a delay in compartmentalisation, suggesting the existence of a compensatory mechanism. Additionally, ESCRT plays a role in nuclear envelope homeostasis (and NPC quality control in yeast) during interphase, but whether this pathway operates through the cell division pathway is unknown. For these reasons, the manuscript presented is interesting, as it tries to separate interphase and mitotic effects of the Lem2/Cmp7 axis.

However, I found the manuscript a too observational and both mechanistically underdeveloped and over-interpreted toward the end, with some strange structuring whereby some figures were only mentioned in the discussion. The manuscript is also very reliant upon interpretation of function from the viability of multiply-deleted and rescued yeast strains, which (given the plethora of functions ascribed to ESCRT proteins, Lem proteins and Lnp1) makes data interpretation confusing.

(Response) We appreciate the valuable comments. According to these comments, we have reorganized the manuscript.

We moved former Supplementary Fig. 8 to Fig. 7 in Results. To clarify the interpretation, we are now focusing on Lem2 and Lnp1. Accordingly, we changed the title. New title:

Lem2 and Lnp1 maintain the membrane boundary between the nuclear envelope and endoplasmic reticulum.

The 1st half of the manuscript, in which the authors build upon their 2016 publication was generally convincing; the excessive membrane expansion and compartmentalisation defect in the lem2Δ lnp1Δ is clear, although how these two proteins (one regulating 3-way junction formation in the peripheral ER and the other regulating INM biology) work together to suppress these problems is unclear. Control movies would be helpful.

(Response) Thank you for this comment. To clarify the phenotype of *lem2Δ lnp1Δ*, we added control movies showing the behavior of Ish1-mCherry in wild type (Movie 2), *lem2Δ* (Movie 3), and *lnp1Δ* (Movie 4). Please see the Supplementary movies. We added the following statements in the text (lines 159-162): **This transient leakage repeatedly occurred at random timings in each individual cell (see individual traces in Fig. 2g; Movie 1). Such transient leakage was not observed in WT, *lem2Δ*, and *lnp1Δ* cells (Supplementary Fig. 4, Movies 2-4).**

The defects in at the ultrastructural level are generally presented clearly, however, methods should be provided for the EM, rather than referencing their 2019 JCS publication, which in turn references their 2010 publication. What are all the white (unstained) regions – are these fixation artefacts? I'm surprised too at the presence of endosomes/vacuoles within the nucleus. As serial sections were taken, can you exclude that these are cytoplasmic regions that are enveloped by the nucleus? Are the 'crystal structure' that you refer to really just a fixation artefact? I think someone with more experience of EM than I ought to look at these.

(Response) According to this comment, we added the method for CLEM analysis in Methods section as follows (lines 522-532): **CLEM imaging was performed --- and analyzed using a transmission electron microscope (JEM1400Plus, JEOL, Japan) at 80 kV."**

We also added EM images of wild type cells as a control in Figure 3e-f, and added statements and quantifications in the text (lines 181-185): **In addition, vacuole-like structures were often observed inside the nucleus (25/49 cells, Fig. 3a, c; arrowheads). Serial section images of the nucleus sometimes showed a crystal-like structure penetrating the nuclear membranes and fused with the vacuolar structures (Fig. 3a and**

d; double arrowheads). Such abnormal membrane structures were not observed in WT cells (0/89 cells, Fig. 3e and f).

The genetic interactions between Lem2's luminal domain and Lnp1 are clear from the data presented in Fig4, but unclear from a topological perspective. In discussion of this figure, it may be prudent to note that the Cmp7 interacting region of Lem2 (its MSC domain) is dispensable for rescuing the growth defect induced by Inp1 loss.

(Response) Thank you for pointing this out. During the process of revision, we have focussed on Apq12 and Vps4, which affect the phenotypes of mutants of *lem2* and *Inp1*. Because deletion of *cmp7*⁺ caused no nuclear protein leakage or growth defects in *lem2ΔInp1Δ* cells, we largely removed the statements related to Cmp7.

When rescuing the growth defect in Inp1d, lem2d, the authors examined localisation of the rescue construct, however, it wasn't clear was whether membrane expansion in the Inp1d, lem2d was the cause of the viability defect? Does Lnp1 d177-231 (4C) rescue the growth defect in 4B?

(Response) According to this comment, we carried out new experiments as follows: we examined whether Lnp1Δ177-231 rescue the growth defect of *lem2Δ Inp1Δ* cells, and found that Lnp1Δ177-231 did not rescue the growth. We added this result in Fig. 4b and in the text (lines 219-221): **Expression of the fragments lacking lunapark domains (Fig. 4b, “+Δ104-334”, “+Δ177-334” and “+Δ177-231”) in *lem2ΔInp1Δ* cells conferred negligible growth recovery.**

Also, are the Lnp1 fragments that localise to 'vacuole' structures really vacuoles, or are they extensions of the ER?

(Response) According to this comment, we stained cells with FM4-64 as a marker for vacuole membranes, and confirmed that Lnp1 fragments (Δ104-334, Δ177-334, Δ177-231) localized to vacuoles. We added this result in Supplementary Fig. 6 and in the text (lines 229-234): **Lnp1 fragments lacking the lunapark domain dispersed through vacuoles as confirmed by co-staining with vacuole membrane-staining reagent FM4-64³⁸ (Fig. 4c, “Δ104-334”, “Δ177-334” and “Δ177-231” and Supplementary Fig. 6). In contrast, the Lnp1 fragment bearing the lunapark domain (“Δ232-334”) recovered its localization to the cortical and perinuclear ER (Fig. 4c).**

An overexpression screen to rescue the growth defect in lem2d Inp1d identified Apq12, previously identified by the Lusk/Kolling as genetically interacting with Cmp7.

Restoration of membrane proliferation and NE permeability in Apq12+ was clear and the authors conclude that Apq12 functions in the Lnp1, but not the Lem2 pathway (L238/9). Further viability assays concern the ability of overexpressed Vps4 and Cmp7 to rescue the growth defect in a lem2Δ, lnp1-N strain, which displays a weak growth defect. Vps4 was able to do this, but I think the conclusion that this means that Vps4 functions were downregulated in the absence of lem2 and lnp1 is an overinterpretation. Does Vps4 overexpression rescue a Lem2Δ lnp1Δ?

(Response 6) According to this comment, we carried out a new experiment to examine whether Vps4 overexpression rescue *lem2Δ lnp1Δ*, and confirmed it. We added this result in Supplementary Figure 7c and in the text (lines 287-288): **We confirmed that overexpression of vps4⁺ also restored the growth of lem2Δlnp1Δ cells (Supplementary Fig. 7c).**

The authors find that Vps4-mCh, but not Cmp7-GFP, is destabilised in lem2Δ and lem2Δ lnp1Δ cells. Is endogenous Vps4 similarly affected?

(Response) We agree that this is an important point. We couldn't compare protein stability of endogenous and fluorescently-tagged Vps4 because we have no anti-Vps4 antibody. However, we believe that endogenous and fluorescently-tagged Vps4 show similar properties because fluorescently-tagged Vps4 is functional in *vps4*-deleted cells. To clarify this point, we added the following statements in the text (lines 296-299): **we confirmed that Vps4-mCherry was functional by performing the following experiments:(1) substitution of endogenous Vps4 with Vps4-mCherry rescued the growth defect of vps4Δ cells; (2) overexpression of Vps4-mCherry rescued the growth defect of lem2Δ lnp1Δ cells (Fig. 6b; Supplementary Fig. 7c).**

This is an interesting result, but I think the authors need to demonstrate why Vps4 is destabilised in the absence of Lem2. The authors quantify the number of Vps4+ endosomes in their cells, and find no difference in the absence of Lem2. Is there a difference in the cytosolic pool? Loss of Lem2 and Lnp1 also destabilises Vps4, but also destabilises Cmp7 now, and reduces the endosomal pool of Vps4. I find this figure largely observational and the mechanism/biology underlying these observations is unclear. Currently, the destabilisation of Vps4 (Lem2-dependent) and the reduction in Vps4+ endosomes (Lem2 and Lnp1 dependent) are not linked. Why is the endosomal pool of Vps4 lost in the double deletion? How are these observations related to the growth defect and membrane expansion (which appeared independent of the Lem2-Cmp7 axis; 4A)?

(Response) We agree that underlying mechanisms are unclear so we removed the related statements in the text. Instead, we added statements on this issue in Discussion as follows (lines 392-400): *The phenotypes observed in lem2Δlnp1Δ cells are remarkably similar to those in vps4Δ cells. The deletion of vps4⁺ gene causes a severely disordered nuclear membrane phenotype and strong growth defect in S. pombe*⁵. It has also been reported that ESCRT-III filament and Vps4 cooperatively remodel the membrane and induce membrane scission by using ATPase activity of Vps4^{42, 43}. Our findings (Fig. 6a, Fig. 6e, and Supplementary Fig. 11) consistently indicate that Lem2 and Lnp1 cooperatively regulate Vps4 localization. Thus, it is likely that Vps4 dysfunctions in lem2Δlnp1Δ cells, and consequently lem2Δlnp1Δ cells exhibit phenotypes similar to vps4Δ cells, that is, abnormal membrane and growth defect.

Is the number of endosomes reduced in lem2Δ lnp1Δ, or just the number of Vps4+ endosomes and is this related to viability/proliferation? I'm afraid I find the observations in this figure unlinked and confusing.

(Response) We appreciate the reviewer for pointing out this important question. To address this question, we carried out new experiments to observe the localization of other ESCRT-III components (Did2, Did4, Vps24, Vps32 and Vps60) in lem2Δ lnp1Δ cells. The result showed that these components remained at endosomal dots and the number of all these dots remained unchanged in lem2Δ lnp1Δ cells, suggesting that number of endosomes was not reduced. Therefore, we concluded that the number of Vps4+ endosomes, but not the number of endosomes, was reduced. We added this result in Supplementary Fig. 11, and added the following statements in the text (lines 318-323): *The number of Cmp7 dots significantly decreased in lem2Δ and lem2Δlnp1Δ cells (Fig. 6h). Other ESCRT-III components (Did2, Did4, Vps24, Vps32, and Vps60) remained at the cytoplasmic dots in lem2Δ, lnp1Δ, and lem2Δlnp1Δ cells (Supplementary Fig. 11). These results suggest that Lem2 and Lnp1 are necessary for the localization of Vps4 at the endosome.*

The authors suggest that loss of Vps4 may hyperactivate ESCRT-III (L273), but I don't think that this interpretation is consistent with our understanding of how ESCRT-III functions; loss of Vps4 is likely to suppress ESCRT-III activity. The triple depletion of lem2, lnp1 and cmp7 is even more toxic than lem2Δ lnp1Δ, (which should be noted in the text) but the membrane proliferation phenotype is not as strong. This makes me question how related the membrane proliferation and viability phenotypes are. I don't

think the fact that Cmp7 deletion failed to rescue the growth of lem2d, Inp1d is evidence that Cmp7 functions independently of Lem2 or Lnp1 (L280, L281).

(Response) Thank you for pointing out this problem. Important message of this experiment is that *cmp7*Δ did not restore the membrane expansion phenotype. The importance of this result became relatively low during the process of revision. Thus, we moved the related figure (former Fig. 7) to Supplementary Fig. 12. In addition, we rephrased the related statements to clarify the point as follows (lines 325-332): *It has been previously reported that depletion of Vps4 induces a strong deformation in the membrane phenotype and growth defect, and these phenotypes are rescued by disruption of the cmp7⁺ gene⁵. Because vps4Δ cells exhibit phenotypes similar to those observed in lem2ΔInp1Δ cells, we speculated that disruption of the cmp7⁺ gene may bypass the phenotypes observed in lem2ΔInp1Δ cells by analogy with vps4Δ cells. To test this possibility, we disrupted the cmp7⁺ gene with lem2⁺ and Inp1⁺. However, cmp7⁺ gene disruption did not rescue the growth defect or the disordered membrane phenotype in lem2ΔInp1Δ cells (Supplementary Fig. 12, lem2ΔInp1Δcmp7Δ).*

We also carried out a new experiment to support the idea that Cmp7 functions independently of Lem2 or Lnp1, and added this result in Supplementary Fig. 13 and the statements in the text as follows (lines 333-341): *Next, we tested whether the nuclear barrier function was defective in cmp7Δ cells. For this purpose, we observed nuclear localization of GFP-GST-NLS during the cell cycle (Supplementary Fig.13). Results showed that no obvious nuclear protein leakage was observed in cmp7Δ cells throughout the cell cycle, unlike in lem2ΔInp1Δ cells (Supplementary Fig. 13; compare with Fig. 2h), although a slight leakage was observed at the end of mitosis (see 20 min after the nuclear division in Supplementary Fig. 13). Thus, it is unlikely that the nuclear protein leakage in lem2ΔInp1Δ cells results from the dysfunction of Cmp7. Taken together, these results indicate that Cmp7 did not bypass the functions of Lem2 and Lnp1.*

The authors next turn to the rescue of viability and membrane hyperproliferation in the Apq12 overexpressing lem2d Inp1d yeast and find that this recovers both the endosomal pool of Vps4 and the expression levels of Vps4, but not those of Cmp7. How this pool is recovered is unclear. Is the destabilisation of Vps4 in lem2d similarly recovered by Apq12 overexpression?

(Response) We appreciate this suggestion. According to this suggestion, we added new Fig. 8d showing protein levels in mutants. In this experiment, we overexpressed Apq12

in *lem2Δ* cells and observed the amount of Cmp7-GFP and Vps4-mCherry. The result showed that Apq12 overexpression recovered the protein level of Vps4 in *lem2Δ* cells as well as *lem2Δ lnp1Δ* cells. We also added statements in the text as follows (lines 363-370): *Western blotting showed that overexpression of Apq12 restored the protein levels of Vps4, which was decreased in lem2Δ and lem2Δlnp1Δ cells to a comparable level of WT cells (Fig. 8d). Thus, Apq12 is likely to stabilize Vps4 and maintain its proper localization during interphase. In contrast, during mitosis, Apq12 overexpression did not restore the SPB localization of Vps4 in lem2Δlnp1Δ cells (Fig. 8e). We also tested the effect of Apq12 overexpression on Cmp7, and found no effects on its localization and protein levels (Fig. 8a, c, d). These results suggest that Apq12 has overlapping functions with Lnp1 in Vps4 localization and stability.*

The data in Figure S8 is not discussed in the results section, but relates to Lem2-dependent SPB localisation of Cmp7 and Vps4 that is not rescued by Apq12 overexpression.

(Response) Thank you for the comment. According to this comment, we moved the former Supplementary Fig. 8 to main Fig. 7, and described these results in the Results section as follows (lines 344-354): *We next examined localization of Vps4 during mitosis because Vps4, together with Cmp7, is predicted to participate in repairing the NE hole at the site where the SPB is inserted and extruded. ---- Considering that the localization of Vps4 was independent of Lem2 during interphase (see Fig. 6c-f), this result suggests that the mitotic localization of Vps4 is regulated differently from that of the interphase.*

I'm afraid I found the ESCRT-part of the manuscript confusing, observational and unlinked and fail to extract a clear message from the data. I could see no data in the manuscript supporting the conclusion (in the title) that Lem2 and Lnp1 act to load Vps4 onto ESCRT-III. The discussion brings this 'loading' conclusion in, but this is at odds with the mechanism by which Vps4 is recruited by ESCRT-III proteins; all ESCRT-III members have MIMs to recruit Vps4 directly and any role for Lem2 or Lnp1 in 'loading' ESCRT-III with Vps4 is confusing.

(Response) We appreciate this comment. We rephrased statements related to the mechanism to regulate Vps4 function throughout the manuscript. We also changed the title to *Lem2 and Lnp1 maintain the membrane boundary between the nuclear envelope and endoplasmic reticulum.*

The discussion is very overinterpreted/speculative, especially from L362 to the end.

(Response) According to this comment, we changed the last paragraph of the discussion (lines 425-437).

N numbers of independent experiments should be provided for all panels.

(Response) According to this comment, we added N numbers of independent experiments in legends or panels.

Reviewer #2

Hirano et al focus on describing the role of three nuclear associated proteins lem2, lnp1, and apq12 in regulating nuclear envelope resealing. While lem2 and apq12 have been previously implicated in an ESCRT-dependent nuclear envelope resealing process, the authors suggest that there is a separate nuclear sealing/repair/maintenance mechanism that is Cmp7-independent. While this study provides very little mechanistic insight into ESCRT-mediated resealing, the Cmp7-independent mechanism is intriguing.

This study would need to go much further to substantiate their model (Figure 9 right side), however, or the authors must remove any mechanistic model prior to publication consideration.

(Response) We greatly appreciate the constructive and valuable comments. We have revised the manuscript according to the comments as described below.

Major points:

1) The Cmp7-independent model in Figure 9 is not plausible. How would ESCRT-III components and Vps4 access the ER-lumen? These proteins are synthesized in the cytosol and do not possess secretion signals that would allow for co-translational folding into the ER lumen and hence access the intramembrane nuclear space. This model must be removed or revised. I suggest that simply the bottom text be included since there is genetic evidence for a Cmp7-independent pathway.

(Response) We deeply appreciate this suggestion. According to this suggestion, we removed mechanistic models in the former Fig. 9. Instead, we presented a short summary of our results in Fig. 8f, and added related statements in the text as follows (lines 374-380): **Our findings suggest that genetic interaction of membrane proteins maintains the membrane boundary between NE and ER, as depicted in Fig. 8f. Lem2 and Lnp1 have a redundant function required for proper Vps4 localization. Apq12 has an overlapping function with Lnp1. Because Vps4 is necessary for membrane remodeling^{44, 45}, Lem2 and Lnp1/Apq12 cooperatively maintain the membrane boundary**

between NE and ER by remodeling membranes through the functions of Vps4 and the ESCRT-III complex independently of Cmp7.

2) There is no microscopic evidence that other ESCRT-III components are found on the surface of the nuclear envelope. The authors appear to loosely connect Vps4 foci to the other ESCRT-III components, however, there is no evidence that the colocalization between ESCRT-III and Vps4 foci (shown in SFig. 7) are residing on the nuclear envelope. These foci could represent the well characterized endosomal/multi-vesicular body functions of ESCRT-III and Vps4. Additionally, there is no evidence in Figure 6 that ESCRT-III foci are enriched on the nuclear membrane in the Cmp7 null background as is suggested by the mechanism in Figure 9. Further, do ESCRT-III foci become enriched at the nuclear envelope in the lem2/Inp1 null background? The statement on lines 251-253 are incorrect given the data.

(Response) We agree that there is no clear evidence that ESCRT-III is enriched on the nuclear envelope in *lem2Δ Inp1Δ* in *S. pombe*. Thus, we removed models in the former Figure 9 and related statements.

3) Lines 124-126 and Fig. 1e: The localization of Rtn1 is shown only in lem2ΔInp1Δ cells, with no WT control. To support the assertion that “tubular ER formation seemed to be disorganized in this mutant”, the localization of Rtn1 in WT cells should be shown for comparison, so that the reader can see how its organization in the mutant cells is abnormal. WT control images are also missing for Rna1 localization (Line 142-144, Fig 2g). These comparisons would be useful for a general cell biology audience.

(Response) Thank you for this suggestion. According to this suggestion, we carried out new experiments, in which we expressed Rtn1-GFP in WT, *lem2Δ* and *Inp1Δ* cells and observed its localization for comparison. We added these results in Fig. 1e.

4) The frequency of Cmp7-GFP foci in Figure 6 and Figure 8 have not been quantified as they were for Vps4.

(Response) Thank you for your suggestion. According to this comment, we quantified the number of Cmp7-GFP foci, added these results in Fig. 6h and Fig. 8c.

5) Figure 2f is not apparently replicated (bottom graph), can this be reproduced with similar magnitudes for fold GFP enrichment to assess error and/or biological variability?

(Response) According to this comment, we show 7 different examples, in which the nuclear protein leakage occurs at different timings (Fig. 2g) in addition to the averaged plot (Fig. 2h).

Minor points:

1) The authors need to justify the statement 279-280 because it seems to be at odds with Gu, M. et al PNAS 2017. Knock-out of lem2 alone prevents CHMP7 localization.

(Response) To clarify this point, we changed the description as follow (lines 325-341):

It has been previously reported that depletion of Vps4 induces a strong deformation in the membrane phenotype and growth defect, and these phenotypes are rescued by disruption of the *cmp7⁺* gene (Gu et al., 2017). --- Thus, it is unlikely that the nuclear protein leakage in *lem2Δlnp1Δ* cells results from the dysfunction of Cmp7. Taken together, these results indicate that Cmp7 did not bypass the functions of Lem2 and Lnp1.

2) The authors use a number of ESCRT-III fluorescent protein fusions including Cmp7-GFP. While it appears that the S. Pombe field is using these ESCRT-III fluorescent protein fusions routinely, this reviewer cannot find any validation study of these probes. Is the mere survival of these organisms carrying fluorescent protein fusions sufficient evidence to prove that ESCRT-mediated functions are not impaired by the probe? This should at least be discussed.

(Response) We appreciate this comment. We moved the most portions of ESCRT-III results to Supplement. We confirmed the functionality of Vps4-mCherry as Vps4 is an essential protein. We added the following statements in the text (lines 296-299): **we confirmed that Vps4-mCherry was functional by performing the following experiments:(1) substitution of endogenous Vps4 with Vps4-mCherry rescued the growth defect of *vps4Δ* cells; (2) overexpression of Vps4-mCherry rescued the growth defect of *lem2Δ Inp1Δ* cells (Fig. 6b; Supplementary Fig. 7c).**

3) SPB was not defined in the manuscript.

(Response) We spelled it out (line 62).

4) Line 77, the word “rapture” is used to describe the nuclear envelope. I believe the authors meant to use the word “rupture”. This is also in Fig 9.

(Response) Thank you for pointing out our mistake. We corrected it (line 66). Fig. 9 was deleted.

5) Lines 137-139 and Fig. 2f: N-values or some measure of the uncertainties should be included for the percentages of WT and single mutant cells exhibiting GFP-GST-NLS leakage.

(Response) According to this comment, we provided the number of cells tested in parentheses, as described in the text as follows (lines 146-149): Leakage of the GFP signal was observed during both interphase and mitosis. The percentage of cells with leakage during mitosis was 97% (143/147) in *lem2Δlnp1Δ* cells, while it was 0.46% (1/216) in WT, 1.1% (2/182) in *lem2Δ*, and 3.0% (7/233) in *lnp1Δ* cells.

6) Fig. 6a, It's not clear why *brr6+* was included in the experiment, or what to make of the data indicating that it further inhibited growth of the mutant cells rather than rescuing. *Brr6* was not discussed in the text except that some of the titles on the reference list suggest that *Brr6* forms a complex with *Apq12*, but I don't think the reader should have to look to the reference list to get a clue as to why this protein might be relevant.

(Response) Thank you for pointing out this. According to this comment, we explained why we used *Brr6* here in the text as follows (lines 284-285): Since *BRR6* has been identified as a genetic interactor of *APQ12* in *S. cerevisiae*, we also tested the overexpression of *brr6+* in *lem2Δ lnp1-ΔN* cells.

Reviewer #3:

In this manuscript Hirano et al, characterize the phenotypes associated to deletion of the nuclear membrane LEM domain containing protein Lem2 and the ER protein Lnp1. They find that both mutants show aberrant NPC-less membrane aggregations and expansions with mixing of ER and NE membranes. This phenotype is exacerbated in the double mutant lem2D lnp1D and is accompanied by loss of nuclear compartmentalization that affects cell viability. Based on this the authors suggest that both proteins function in maintaining the ER/NE membrane compartmentalization. This conclusion is supported with nice CLEM images of these mutants where overdeveloped membrane structures and NE breakage are observed. They further identify Lnp1 and Lem2 functional domains by analysing the extent of suppression of these aberrant membrane phenotypes of the lem2Dlnp1D mutant by different Lnp1 and Lem2 protein fragments.

They found that the ER protein apq12 and vps4, a component of the ESCRT-III membrane remodelling system, are multi-copy suppressor of the growth and membrane

phenotypes of the absence of Lem2 and Lnp1 function, and apq12 in addition also recovers nuclear compartmentalization during interphase but not during mitosis. These results and also the strong genetic interaction between lem2D apq12D and lnp1Dlem2D but not in lnp1Dapq12D support the hypothesis that that apq12 and lnp1 function in the same pathway at the ER/NE different from the Lem2 pathway.

They further explore the relationship between these membrane regulators and the conserved ESCRT-III/Vps4 system that is required for proper membrane sealing and reformation and that has been poorly characterized in the fission yeast. They find that ESCRT-III is indeed regulated by two partially redundant pathways. Lem2 and Lnp1/Apq12 are required for proper localization of Vps4 to endomembranes, whereas exclusively Lem2 is required for proper localization of the ESCRT-III adaptor Cmp7 to the NE at the SPBs during mitosis. This Cmp7 dependent function of ESCRT-III seems to be especially important during mitosis, where it is required to avoid nuclear leakage at time of SPB extrusion from the NE

The results presented here are convincing and the message of this study is interesting as it highlights important differences in the spatial regulation of a conserved membrane remodelling system. The distinction between ESCRT-III regulation at the NE and at other compartments of the endomembrane system, is of great interest in the field and the role of NE and ER proteins that regulate membrane structure and lipid composition in these processes reinforces a new and interesting role of different lipid domains in a variety of nuclear functions as has been demonstrated by the same authors in previous studies (Kinugasa et al, 2019).

So I am in favour of its publication but I still have some points that might be clarified:

We greatly appreciate the constructive and encouraging comments.

1.Introduction, lines 54-62.

It is inferred here that the ESCRT-III system coupled with Spastin is a conserved mechanism that operates to seal any membrane hole in S.cerevisiae, S.pombe and metazoans. To my knowledge this system (at least Spastin function) has only been described in human cells and not in yeasts and only in sealing those remaining holes that remain in contact with spindle MTs. I think this point might be easily clarified.

(Response) Thank you for this reasonable comment. During the process of extensive revisions, these statements have been removed. Thank you again.

2.Figure 1e. There is only one panel showing Rtn1 localization in the double mutant. The authors should include a wt as a control as the wt localization of Rtn1has not been

previously shown in the manuscript (in fact in the figure legend it is included the wt strain).

(Response) Thank you for this suggestion. According to the comment, we carried out new experiments, in which we expressed Rtn1-GFP in WT, *lem2Δ* and *Inp1Δ* cells and observed its localization for comparison. We added these results in Fig. 1e.

3. Page 6 line 131-132. "with severe NE and ER membrane defects resulting in breakage of the nuclear sealing and nuclear transport defect." There is no data to this point that supports a defect in nuclear transport, at least in NPC-dependent transport, indeed NPCs localize properly. (See below, point 6)

(Response) We agree that there is no data to support a defect in nuclear transport. So we rephased the leading sentence of this paragraph as follows (lines 139-142):

*To examine whether the severe deformations of the NE and ER membranes in *lem2ΔInp1Δ* cells are associated with breakage of the nuclear barrier function, we observed localization of the nuclear proteins by expressing nuclear localization signal (NLS) tagged with GFP-GST (GFP-GST-NLS).*

And we concluded this paragraph with the following sentence (lines 165-167):

*Because the leakage of the GFP signal was transient and the signal was able to recover in the nucleus during a relatively short period of time, it is unlikely that nuclear transport is defective in *lem2ΔInp1Δ* cells.*

4a. Figure 2d. The mitotic leakage is very penetrant (97% of mitotic events). I wonder whether the loss of compartmentalization shown in the double mutant in fig 2 d in interphase, is the result of a defect in NE remodelling that occurs during mitosis (as shown in f) likely during SPB extrusion from the NE or whether there are other events during interphase such as SPB insertion that contribute to this phenotype. Can the authors show the frequency of interphase leakage? Can the authors find a correlation between the position and/or timing of SPB insertion and nuclear leakage?

(Response) Thank you for raising this important question. To address this question, we carried out a new experiment, in which we expressed Atb2-mCherry in GFP-GST-NLS-expressing cells to monitor the mitotic process. We added the obtained result in Fig. 2f and g, and added the following statements in the text (lines 149-155).

We also presented movies showing the nuclear leakage (Supplementary Movie 1-4).

To dissect the timing of the mitotic leakage, we co-expressed Atb2-mCherry to visualize the mitotic spindle simultaneously with GFP-GST-NLS (Fig. 2f and Supplementary Fig.

4). In *lem2ΔInp1Δ* cells undergoing mitosis, leakage started immediately after mitotic spindle formation, reached a maximum during chromosome segregation, and continued until mitotic spindle disassembly (Fig. 2f; see Fig. 2g, h for quantification), suggesting that the mitotic leakage correlates with SPB insertion/extrusion.

4b. The fact that overexpression of Apq12 rescues the membrane defects of the lem2Inp1 mutant during interphase but not during mitosis is striking and suggest that indeed different mechanism regulate NE maintenance during interphase and during mitosis. Additionally, different components of the membrane system might be preferentially remodelled during interphase or during mitosis.

(Response) Thank you for this important comment. To address these points, we carried out live cell imaging of Apq12-expressing *lem2Δ Inp1Δ* cells, and confirmed that the leakage phenotype during mitosis is not restored by Apq12 expression. We added these results in Supplementary Figure 9 and Movie 5, and added the following statements in the text (lines 261-265): **Under these conditions, expression of Apq12 rescued the NE and ER membrane defects during interphase, that is, abnormal NE shape (Fig. 5b and c) and nuclear protein leakage (Fig. 5d and e). However, the defect in nuclear protein leakage during mitosis was not restored (arrows in Fig. 5d; Supplementary Fig. 9 and Movie 5).**

5. Figure 2 g / page 7, lines 142-143.

This experiment is confusing to me. If I am not mistaken, in a previous study, the same authors showed that Rna1 is cytoplasmic but once it enters the nucleus (by forcing its localization during interphase) it leads to a sort of "virtual nuclear envelope break down" (Asakawa et al, 2010) and the release of nuclear components. According to this, Rna1 entrance in the nucleus would promote the further release of nuclear components. I think that the GFP-GST-NLS or RFP-NLS are good markers to follow nuclear compartmentalization and that the result on Rna1 might distract from the important points of this study. In addition this experiment is not explained nor discussed in the main text so I suggest removing this experiment.

(Response) Thank you for this comment. According to the suggestion, we removed this result.

6. page7 lines 156-158. Here again authors suggest that the loss of nuclear barrier function in the double mutant is caused by nuclear membrane rupture rather than transport defects or the NPC, but they do not discuss the reasons for this conclusion.

The localization of GFP-GST-NLS (Figure 2F), and RFP-NLS (Figure 2g) in the nucleus and the reimport of these proteins once nuclear compartmentalization is recovered in the double mutant prove that NPCs are functional. I would make this point more explicit (maybe moving the conclusion in lines 156-158 to line 143).

(Response) Thank you for this comment. We made explicit statements in lines 165-167 as follows: **Because the leakage of the GFP signal was transient and the signal was able to recover in the nucleus during a relatively short period of time, it is unlikely that nuclear transport is defective in *lem2Δlnp1Δ* cells.**

7. page 12 lines 279-280. "therefore, Cmp7 seems to function through a pathway independent of Lem2 or Lnp1". I think that this conclusion is not supported by the data showing that Cmp7 localization is strongly dependent on Lem2. Have the authors looked specifically whether deletion of Cmp7 affects the mitotic loss of compartmentalization? I understand from the results that the aberrant global nuclear membrane phenotypes might be independent on the specific loss of compartmentalization that occurs during mitosis (during SPB externalization for example) as this is not rescued by overexpression of Apq12 (Fig 5d).

(Response) We appreciate this insightful comment. According to this comment, we observed a behavior of GFP-GST-NLS in *cmp7Δ* cells during mitosis. Results obtained supported our statement that Cmp7 function through a pathway independent of Lem2 and Lnp1. To clarify the point raised by this reviewer, we added this result in Supplementary Figure 13, and added the following statements in the text (lines 333-341): **Next, we tested whether the nuclear barrier function was defective in *cmp7Δ* cells. For this purpose, we observed nuclear localization of GFP-GST-NLS during the cell cycle (Supplementary Fig.13). Results showed that no obvious nuclear protein leakage was observed in *cmp7Δ* cells throughout the cell cycle, unlike in *lem2Δlnp1Δ* cells (Supplementary Fig. 13; compare with Fig. 2h), although a slight leakage was observed at the end of mitosis (see 20 min after the nuclear division in Supplementary Fig. 13). Thus, it is unlikely that the nuclear protein leakage in *lem2Δlnp1Δ* cells results from the dysfunction of Cmp7. Taken together, these results indicate that Cmp7 did not bypass the functions of Lem2 and Lnp1.**

8. Finally, the authors propose that these functions of Lem2, Lnp1 and Apq12 might be achieved through changes in membrane structure and/or lipid composition. In this sense, the same authors have shown in a previous study that the overexpression of the fatty acid elongase Elo2 rescues most of Lem2 described phenotypes and also the

more penetrant NE aberrant phenotypes and loss of compartmentalization during mitosis of the double mutant lem2bqt4. I wonder whether overexpression of Elo2 would rescue the nuclear compartmentalization of lem2Dlnp1D specifically during mitosis (and perhaps Cmp7 localization)

(Response) We are happy to discuss this point. We also have a great interest whether Elo2 rescue the loss of compartmentalization in *lem2Δ Inp1Δ* cells. We are continuing extensive studies on the effect of Elo2 on a series of NE mutants, so we would like to address the effect of Elo2 in future in a more complete form.

Minor comments:

-Title: ER should not be abbreviated.

(Response) Following this suggestion, we spelled it out (line 4).

-Page 4 line 77: "rapture" should be "rupture".

(Response) Thank you for pointing out our mistake. We corrected it (line 66).

- I am concerned about the size of some of the figures, especially when they are printed. As examples, fig 1c and d left panels. Figure S2 a, b and c left panels, and figure S8.

(Response) According to this suggestion, we made them bigger and checked that they are big enough to see after printing.

REVIEWERS' COMMENTS:

Reviewer #1 (Remarks to the Author):

I think that with the extensive textual and figure revisions, the authors have done a good job in improving the manuscript. Although it still feels like the ESCRT section is mechanistically underdeveloped, my specific comments have been largely addressed and I think the manuscript is clearer. I'm happy to support publication.

A couple of minor concerns:

L158-159, GFP-GST-NLS leakage was said to occasionally occur in interphase and nuclear compartmentalisation appears functional during interphase in the live imaging in 2F, but in 2D/2E interphase rupture was observed (L145-146). Can these be reconciled?

It is still not clear to me how the luminal domain of LEM2 can function with the cytoplasmic LNP domain of LNP1. Nor is it just the LNP domain that is the necessary part, as you proceed to investigate, the N-terminal region is also needed. I think this statement should be added to the conclusion on L234-L235.

Reviewer #3 (Remarks to the Author):

The authors have made a great effort to satisfy all the reviewers' concerns. I think the manuscript has improved substantially with the revision and I think it is ready for publication.

Response to Reviewers

We greatly appreciate the valuable comments and made changes to the manuscript according to those comments. Point-by-point responses to the comments are described below. *Blue italic sentences are reviewer's comments*, black roman sentences are our responses, and **red fonts are quotation from the text**.

(comment 1) L158-159, GFP-GST-NLS leakage was said to occasionally occur in interphase and nuclear compartmentalisation appears functional during interphase in the live imaging in 2F, but in 2D/2E interphase rupture was observed (L145-146). Can these be reconciled?

(response) Thank you for pointing out this. Transient leakage occurs in *lem2Δ lnp1Δ* at the timing indicated by red arrows in Fig. 2f and g, and it is clear in comparison with wild-type and single mutants, especially in Movies. To clarify this point, we revised it as follows:

(lines 155-159) **"During interphase in *lem2Δ lnp1Δ* cells, the transient leakage of GFP-GST-NLS occasionally occurred (time of leakage indicated by red arrows in Fig. 2f and g). Such transient leakage repeatedly occurred at random timings in each individual cell (see individual traces in Fig. 2g; Movie 1), but not observed in WT, *lem2Δ*, and *lnp1Δ* cells (Supplementary Fig. 4, Movies 2-4)."**

(comment 2) It is still not clear to me how the luminal domain of LEM2 can function with the cytoplasmic LNP domain of LNP1. Nor is it just the LNP domain that is the necessary part, as you proceed to investigate, the N-terminal region is also needed. I think this statement should be added to the conclusion on L234-L235.

(response) Thank you for pointing out this. As suggested, we added the statement describing "the N-terminal region is also needed". To do this, we first concluded the domain necessary for localization, and made a separate paragraph summarizing necessary domains for functions. In the summarizing paragraph, we added "N-terminal and lunapark domains of Lnp1 are necessary" (underlined below). We also added statement suggesting involvement of other factors for functional interaction between Lem2 and Lnp1.

(lines 231-236) **"These results indicate that the conserved lunapark domain is crucial for proper localization of Lnp1.**

Collectively, these domain analyses indicate that the luminal region of Lem2 and the cytoplasmic N-terminal and lunapark domains of Lnp1 are necessary for their genetic interaction although other factors may be involved in connecting Lem2 and Lnp1."